# Splicing variation of BMP2K balances abundance of COPII assemblies and autophagic degradation in erythroid cells

Jaroslaw Cendrowski[1]*, Marta Kaczmarek[1], Michał Mazur[1], Katarzyna Kuzmicz-Kowalska[1], Kamil Jastrzebski[1], Marta Brewinska-Olchowik[2], Agata Kominek[2], Katarzyna Piwocka[2], Marta Miaczynska[1]*

[1]Laboratory of Cell Biology, International Institute of Molecular and Cell Biology, Warsaw, Poland; [2]Laboratory of Cytometry, Nencki Institute of Experimental Biology, Warsaw, Poland

**Abstract** Intracellular transport undergoes remodeling upon cell differentiation, which involves cell type-specific regulators. Bone morphogenetic protein 2-inducible kinase (BMP2K) has been potentially implicated in endocytosis and cell differentiation but its molecular functions remained unknown. We discovered that its longer (L) and shorter (S) splicing variants regulate erythroid differentiation in a manner unexplainable by their involvement in AP-2 adaptor phosphorylation and endocytosis. However, both variants interact with SEC16A and could localize to the juxtanuclear secretory compartment. Variant-specific depletion approach showed that BMP2K isoforms constitute a BMP2K-L/S regulatory system that controls the distribution of SEC16A and SEC24B as well as SEC31A abundance at COPII assemblies. Finally, we found L to promote and S to restrict autophagic degradation and erythroid differentiation. Hence, we propose that BMP2K-L and BMP2K-S differentially regulate abundance and distribution of COPII assemblies as well as autophagy, possibly thereby fine-tuning erythroid differentiation.

*For correspondence:
jcendrowski@iimcb.gov.pl (JC);
miaczynska@iimcb.gov.pl (MM)

**Competing interests:** The authors declare that no competing interests exist.

## Introduction

In eukaryotic cells, vesicular transport underlies endocytosis and exocytosis, ensuring the proper distribution of transmembrane proteins between cellular compartments. It involves vesicles which, depending on their origin, can be formed by one of protein coat assemblies (clathrin, COPI and COPII) (*Gomez-Navarro and Miller, 2016*). Clathrin assembles into lattices shaping vesicles transporting cargo from the plasma membrane (PM) or the *trans*-Golgi network to the endolysosomal system (*Robinson, 2015*). COPI and COPII vesicles transport cargo within the early secretory pathway. COPII vesicles bud at the ER exit sites (ERES) and deliver cargo to the ER-Golgi intermediate compartment (ERGIC) and the Golgi (*Venditti et al., 2014*), while COPI vesicles transport cargo from the Golgi and the ERGIC back to the ER and between the Golgi cisternae (*Arakel and Schwappach, 2018*).

As initially characterized in yeast, COPII–mediated transport involves sequential recruitment of coat components, including the Sar1 GTPase, the Sec23/Sec24, and the Sec13/Sec31 subcomplexes. The Sec23/Sec24 inner shell sorts cargo into ER-derived vesicles while Sec13 and Sec31 polymerize forming their outer cage. The scission of budding vesicles occurs due to Sar1 GTPase activity which is stimulated by the assembled coat, particularly by the recruitment of Sec31 (*Bielli et al., 2005*; *Lee et al., 2005*; *Sato and Nakano, 2005*; *Townley et al., 2008*). COPII vesicle production is regulated by Sec16 protein by two distinct mechanisms, either by providing a scaffold organizing COPII assembly (*Bhattacharyya and Glick, 2007*; *Connerly et al., 2005*; *Ivan et al., 2008*; *Martínez-Menárguez et al., 1999*; *Watson et al., 2006*) or by negative regulation of COPII turnover through

inhibition of the Sec31 recruitment (*Bharucha et al., 2013*; *Kung et al., 2012*; *Yorimitsu and Sato, 2012*).

Vesicular trafficking contributes to autophagosome formation upon induction of macroautophagy (hereafter referred to as autophagy) (*Lamb and Tooze, 2016*). Autophagosome biogenesis is initiated at the ER and occurs via incorporation of vesicles derived from ERES or endosomal recycling compartment (*Farhan et al., 2017*; *Lamb et al., 2016*; *Sanchez-Wandelmer et al., 2015*). In yeast, Sec16, Sec23 and Sec24 are required for autophagosome formation (*Ishihara et al., 2001*), while in mammalian cells ERES contributes to autophagy via a subset of COPII vesicles marked by particular isoforms of the inner shell proteins, SEC23B, SEC24A and SEC24B (*Jeong et al., 2018*).

The membrane transport pathways undergo profound remodeling upon cell differentiation. Particularly rapid and intense membrane rearrangements occur during erythroid cell maturation. In human, erythroid progenitors undergo enormous expansion to fulfill the daily requirement of around $2 \times 10^{11}$ new erythrocytes (*Dzierzak and Philipsen, 2013*). This robust differentiation involves efficient iron uptake through transferrin endocytosis and activation of pathways for organelle removal such as their autophagic clearance (*Moras et al., 2017*; *Ney, 2011*). This needs to be coordinated with a delivery of a vast amount of erythroid-specific surface markers through the secretory pathway (*Satchwell et al., 2011*; *van den Akker et al., 2010*). However, although proper COPII-dependent secretion is indispensable for erythropoiesis (*Bianchi et al., 2009*; *Satchwell et al., 2013*; *Schwarz et al., 2009*), terminal erythroid differentiation is associated with a loss of COPII coat components (*Satchwell et al., 2013*).

A possible candidate protein that could be involved in rearrangement of membrane trafficking pathways during cell differentiation is bone morphogenetic protein 2 (BMP-2)-inducible kinase (BMP2K). It has been discovered as a transcriptional target of BMP-2 in osteoblast differentiation (*Kearns et al., 2001*) but later identified as an interactor of proteins involved in clathrin-mediated endocytosis (CME) (*Borner et al., 2012*; *Brehme et al., 2009*; *Krieger et al., 2013*). Up to date, the cellular function of this member of the Ark1/Prk1 family of serine/threonine kinases is ill-defined and none of its phosphorylation targets are established, although it is suspected to phosphorylate the medium (μ2) adaptin of the AP-2 clathrin adaptor complex (*Wrobel et al., 2019*). Recent reports implicated BMP2K in leukemogenesis (*Tokyo Children's Cancer Study Group (TCCSG) et al., 2017*; *Pandzic et al., 2016*; *Wang et al., 2020*). In a study that did not consider its possible endocytic functions, BMP2K was fished out as a putative stimulator of autophagy, potentially required for erythropoiesis (*Potts et al., 2013*). However, no insights were provided into possible mechanisms of how this putative endocytic kinase would regulate autophagy.

Here we report that splicing variants of BMP2K constitute a two-element system regulating distribution and abundance of COPII assemblies and autophagy. Our study uncovers an unusual mechanism of two splicing variants of a kinase playing opposing roles in intracellular processes that may allow for their fine-tuning during cell differentiation.

## Results

### The levels of BMP2K splicing variants are initially increased but later reduced during mouse erythroid differentiation

To choose a proper biological context to study cellular functions of BMP2K, we mined gene expression databases and found that the expression of human *BMP2K* gene is high in the early erythroid lineage (biogps.org) and upregulated during erythroid maturation in a manner similar to that of erythroid-enriched markers, such as TFRC (transferrin receptor 1) (*Novershtern et al., 2011*). To verify these data, we analyzed mRNA abundance of mouse BMP2K in an ex vivo erythropoiesis model. According to the UniProtKB database, mouse expresses two splicing variants (isoforms) of the kinase, the longer (BMP2K-L) and the shorter (BMP2K-S), which result from alternative mRNA splicing. We observed that in isolated mouse fetal liver erythroblasts differentiated with erythropoietin (EPO)-containing medium, mRNA levels of BMP2K-L and BMP2K-S increased gradually, similarly to TFRC (*Figure 1—figure supplement 1A*).

We next analyzed protein levels of TFRC and BMP2K variants at consecutive time-points (24, 48 and 72 hr) of differentiation. While the amounts of TFRC were markedly elevated, the abundance of BMP2K-L and -S was initially upregulated and subsequently downregulated (*Figure 1A*).

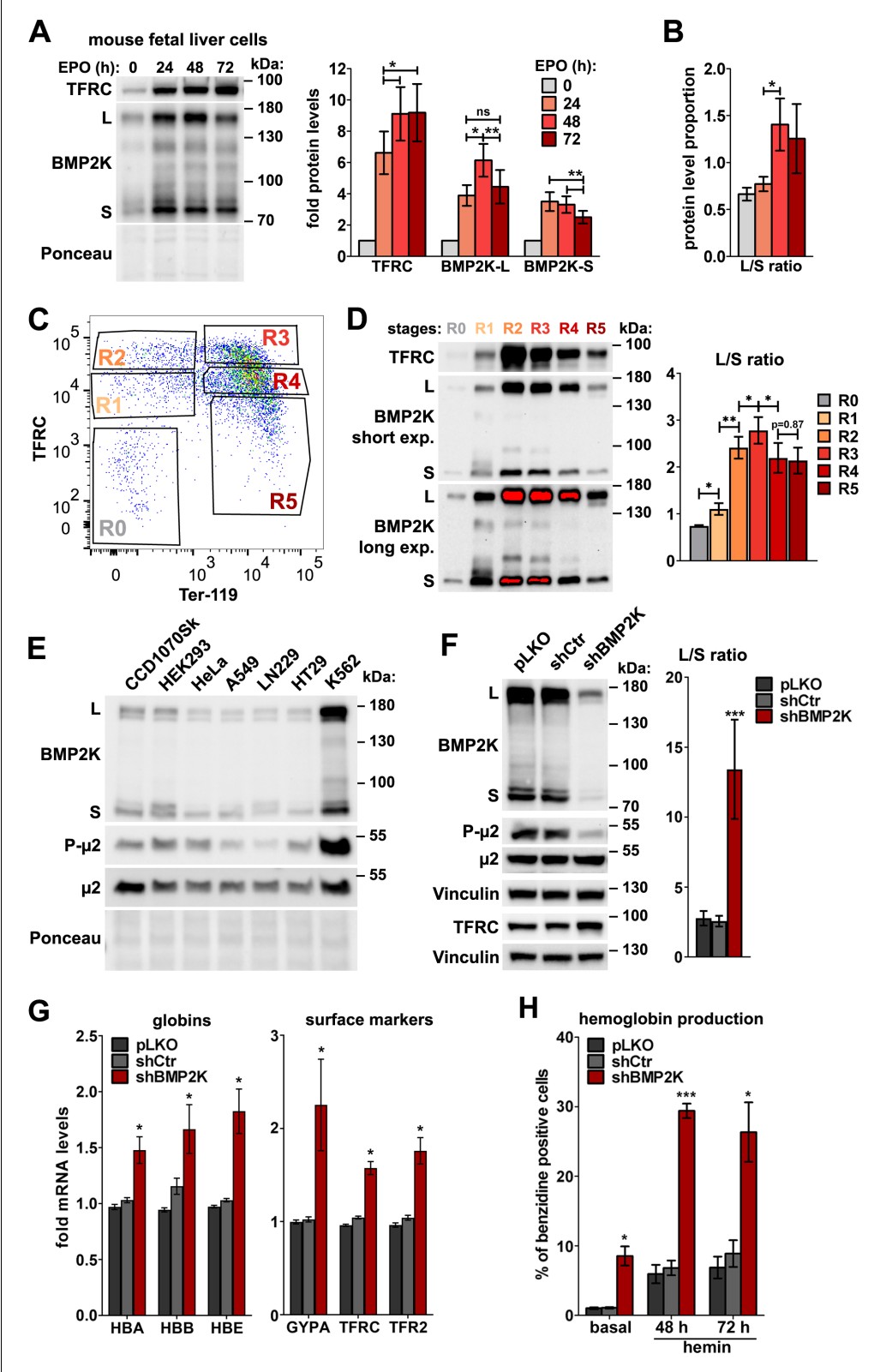

**Figure 1.** In the erythroid cells, BMP2K splicing variants are enriched and their reduction promotes erythroid differentiation. (**A**) Western blots showing the levels of TFRC and BMP2K splicing variants (L and S) at different time-points during erythropoietin (EPO)-stimulated differentiation of mouse fetal liver erythroblasts. Graphs show fold changes in non-normalized protein levels obtained by densitometric analysis of western blotting results (n = 5 +/- SEM). (**B**) The proportion between the detection intensities of BMP2K-L and -S (L/S ratio) calculated after densitometric analysis of bands from western

*Figure 1 continued on next page*

*Figure 1 continued*

blots represented in A (n = 6 +/- SEM). (C) Dot plot showing fluorescence intensities of the indicated markers on the surfaces of mouse fetal liver erythroblasts stimulated with EPO for 96 hr. Gates distinguish consecutive differentiation stages (R0–R5) of erythroblasts isolated by fluorescence activated cell sorting (FACS). (D) Western blots (short and long exposures) showing the levels of TFRC and BMP2K variants in the indicated FACS-isolated differentiation stages of erythroblasts. Graph shows the L/S ratio calculated after densitometric analysis of western blotting results (n = 5 +/- SEM). (E) Western blots showing the levels of BMP2K splicing variants or total and Thr156-phosphorylated μ2 (P-μ2) in lysates from the indicated human cell lines. Ponceau staining serves as a gel loading control. (F) Western blots showing the efficiency of depleting all BMP2K variants using shRNA (shBMP2K) in K562 cells, and its effect on the levels of total and phosphorylated μ2 and TFRC as compared to non-depleted cells (empty pLKO vector or non-targeting shRNA, shCtr). Graph shows the L/S ratio calculated after densitometric analysis of western blotting results (n = 6 +/- SEM). (G) Fold changes in mRNA levels of the indicated erythroid markers in control cells or in cells depleted of all BMP2K splicing variants using shRNA (n = 3 or 4 +/- SEM). (H) Percentage of benzidine-positive control cells or cells depleted of BMP2K using shRNA, under basal growth conditions or after stimulation for 48 hr or 72 hr with 20 μM hemin (n = 3 +/- SEM). Values measured for BMP2K-depleted cells (F, G and H) were compared statistically to those measured for shCtr-treated cells. *p<0.05, **p<0.01, ***p<0.001.

The online version of this article includes the following source data and figure supplement(s) for figure 1:

**Source data 1.** Numerical data for graphs in *Figure 1*.
**Figure supplement 1.** Expression of *BMP2K* gene in erythroid cells and the effect of its CRISPR/Cas9-mediated silencing on erythroid differentiation.
**Figure supplement 1—source data 1.** Numerical data for graphs in *Figure 1—figure supplement 1*.
**Figure supplement 2.** Benzidine staining of K562 cells depleted of all BMP2K splicing variants.
**Figure supplement 3.** The effects of depleting all BMP2K splicing variants on endocytosis.
**Figure supplement 3—source data 1.** Numerical data for graphs in *Figure 1—figure supplement 3*.

Noteworthy, the proportion between the intensities of western blotting detection of the two isoforms (L/S ratio) changed with time of differentiation, as BMP2K-S protein was upregulated earlier (the highest levels detected at 24 hr) than that of BMP2K-L (the highest levels detected at 48 hr) (*Figure 1B*).

EPO-stimulated mouse fetal erythroblast cultures are a heterogeneous mixture of cells at various differentiation stages (*Zhang et al., 2003*). To assess precisely the amounts of BMP2K variants at particular stages, we labelled EPO-stimulated cells with antibodies recognizing mouse erythroid surface markers, CD71/TFRC and Ter-119 (*Zhang et al., 2003*). This allowed us to isolate, by fluorescence activated cell sorting (FACS), the earliest primitive progenitors (CD71$^{low}$/Ter-119$^{low}$ – population R0) as well as further stages of erythroblast differentiation (consecutively: CD71$^{med}$/Ter-119$^{low}$ – R1, CD71$^{high}$/Ter-119$^{low}$ – R2, CD71$^{high}$/Ter-119$^{high}$ – R3, CD71$^{med}$/Ter-119$^{high}$ – R4 and CD71$^{low}$/Ter-119$^{high}$ – R5) (*Figure 1C*). Consistently with the analysis of heterogeneous cultures (*Figure 1B*), the levels of BMP2K isoforms were low in the early stages (R0 and R1), the highest in the transitory stages (R2 and R3) and reduced at the last stages (R4 and R5) (*Figure 1D*). Again, erythroid differentiation was associated with a change in the L/S ratio. It was in favor of BMP2K-S (L/S < 1) in primitive progenitors (R0) but was shifted towards BMP2K-L (L/S > 1) upon differentiation, being the highest at R3 (L/S > 2.5) and remaining high at R4 and R5 (L/S ~ 2) when total BMP2K protein levels were again downregulated (*Figure 1D*).

Thus, we found that during erythroid differentiation, the levels of BMP2K splicing variants are initially upregulated and subsequently reduced. The ratio between the two isoforms changes during erythroid differentiation, with BMP2K-S being predominant in the early erythroid precursors and BMP2K-L prevailing during differentiation and maturation.

## Reducing the levels of BMP2K splicing variants in K562 cells promotes erythroid differentiation

The observed changes in protein abundance of mouse BMP2K variants upon differentiation could suggest that the initial increase of their levels would promote early steps of erythropoiesis while their subsequent decrease would favor erythroid maturation. To verify this complex hypothetical scenario, we sought a simpler cellular model. We found that K562 human erythroleukemia cells contain much higher amounts of BMP2K than various immortalized, solid tumor or non-erythroid blood cancer cell lines (*Figure 1E* and *Figure 1—figure supplement 1B*). In K562 cells the longer isoform (BMP2K-L) appeared as more abundant than the shorter (BMP2K-S) (*Figure 1E,F* and *Figure 1—figure supplement 1B,C*). Their L/S ratio was approximately 2:1 (calculated for control cells shown in *Figure 1F* and *Figure 1—figure supplement 1C*).

K562 cells have erythroid progenitor-like features (*Andersson et al., 1979*) and can initiate erythroid differentiation (*Barbarani et al., 2017*; *Bu et al., 2014*; *Ma et al., 2013*; *Wang et al., 2011*; *Wu et al., 2018*). To learn whether BMP2K silencing would reverse or advance erythroid differentiation of K562 cells, we silenced *BMP2K* gene expression using shRNA (shBMP2K) or CRISPR/Cas9 (gBMP2K#1 or #2) approaches. Although we did not achieve a complete *BMP2K* gene knock-out with the CRISPR/Cas9 approach, using both techniques we obtained efficient reduction of BMP2K variant levels. We noticed that BMP2K depletion was associated with a concomitant upregulation of the control 2:1 L/S ratio, very strongly (up to 13:1) upon shBMP2K and less potently (up to 3:1 or 4:1) upon gBMP2K#1 or #2 (*Figure 1F* and *Figure 1—figure supplement 1C*). A possible cause of differences in the L/S ratio upon various BMP2K depletion techniques remains obscure.

To assess the effects of BMP2K depletion on erythroid differentiation, we measured the expression levels of erythroid-specific genes, by qPCR, and the production of hemoglobin, by benzidine staining. As reported (*Villeval et al., 1983*), control K562 cells expressed several erythroid-specific genes (*Figure 1G* and *Figure 1—figure supplement 1D*) but only 0.5–1% of these cells were positive for hemoglobin (*Figure 1H*, *Figure 1—figure supplements 1E* and *2A,B*). This percentage increased upon treatment with a heme precursor hemin, as described (*Ma et al., 2013*; *Wang et al., 2011*). We found that BMP2K depletion promoted erythroid differentiation of K562 cells. shBMP2K markedly elevated expression of erythroid-specific genes (*Figure 1G*) and potently increased the number of hemoglobin-positive cells under basal culture conditions (by 6-fold) and in the presence of hemin (by 4-fold) (*Figure 1H* and *Figure 1—figure supplement 2A*). In turn, both gRNAs weakly induced the expression of erythroid markers (*Figure 1—figure supplement 1D*) and stimulated hemoglobin production less potently than shBMP2K (~3 fold in basal conditions and ~2 fold upon hemin treatment) (*Figure 1—figure supplements 1E* and *2B*).

Collectively, elevated expression of erythroid markers and increased hemoglobin production in K562 cells depleted of all BMP2K splicing variants suggested that at least one of the variants inhibits cellular events responsible for erythroid differentiation. As the extent of differentiation correlated with the ratio between the remaining L and S levels it is possible that the balance between variant abundance affects erythroid maturation. These observations need to be verified in more physiological models of erythropoiesis.

## The role of BMP2K in CME does not explain its involvement in erythroid differentiation

As BMP2K is a putative endocytic kinase found among interactors of μ2 (*Brehme et al., 2009*), it could affect erythroid differentiation via regulation of AP-2-dependent CME. Consistent with its postulated role in phosphorylating the μ2 adaptin at Thr156 (*Wrobel et al., 2019*), K562 cells had high phospho-μ2 levels (*Figure 1E*) that were strongly reduced upon BMP2K depletion (*Figure 1F* and *Figure 1—figure supplement 1C*). However, while the phosphorylation of μ2 was shown to promote endocytosis (*Olusanya et al., 2001*; *Ricotta et al., 2002*; *Wrobel et al., 2019*), BMP2K depletion increased continuous uptake of fluorescently labelled Tf and 10 kDa dextran (a fluid-phase marker) by K562 cells (*Figure 1—figure supplement 3A–E*). shBMP2K increased early (5 min) and steady state (40 min) Tf uptake as well as dextran internalization, all by around 30% (*Figure 1—figure supplement 3B,C*). gBMP2K#1 or #2 had a weaker effect on endocytosis than shBMP2K, increasing early and steady state Tf uptake roughly by 10–15% and dextran internalization only by 5–15% (*Figure 1—figure supplement 3D,E*).

The elevated Tf uptake despite lower μ2 phosphorylation in BMP2K-depleted cells could occur due to the increase in expression of genes encoding transferrin receptors, as described above (*Figure 1G* and *Figure 1—figure supplement 1D*). Consistently, shBMP2K led to elevated TFRC protein levels (*Figure 1F*) and higher binding of Tf to cell surface on ice (0' in *Figure 1—figure supplement 3F,G*, empty bars), arguing for a higher number of Tf receptor molecules available on the PM. A pulse-chase uptake assay of the pre-bound Tf showed a reduction of Tf internalization efficiency (% of surface-bound Tf that was internalized at 37 °C) in shBMP2K-treated cells (down to 62% from 70% in control cells) (*Figure 1—figure supplement 3H*). However, due to elevated Tf binding, their resultant Tf pulse-chase uptake was higher than in control cells (*Figure 1—figure supplement 3G*).

Collectively, despite lower Tf endocytosis efficiency, cells lacking all BMP2K variants showed higher continuous Tf uptake, possibly due to elevated surface abundance of Tf receptors. The latter

observation made the effects of BMP2K depletion on Tf uptake difficult to dissect. Thus, although we confirmed the implication of BMP2K kinase in CME, this role could not explain its involvement in erythroid differentiation. Therefore, we hypothesized that in addition to endocytosis, BMP2K variants could regulate other cellular events to affect red blood cell maturation.

## BMP2K splicing variants can associate with SEC16A protein and localize to the early secretory compartment

To find out in which other cellular processes the two BMP2K isoforms could function, we investigated their interactomes. To this end, we performed proximity biotinylation (BioID) followed by liquid chromatography coupled to tandem mass spectrometry (LC-MS/MS) in HEK293 cells ectopically expressing the L or the S variant tagged with a mutant BirA biotin ligase (BirA*) at their N- or C-termini (*Figure 2—figure supplement 1A,B*). To avoid artifacts due to BirA tag at either of the termini, we focused on proteins detected as common interactors of N- and C-terminally tagged BMP2K isoforms (*Supplementary file 1*-Table 1,2, *Figure 2A*, *Figure 2—figure supplement 1C*). Within the two variant-specific interactomes, we found proteins involved in vesicular transport or mRNA translation and transport (*Figure 2—figure supplement 1C*).

Among regulators of vesicular transport found as proximal to both BMP2K isoforms were CME adaptors: NUMB, PICALM, EPS15R, and AGFG1 (HRB) (*Benmerah et al., 1995*; *Chaineau et al., 2008*; *Coda et al., 1998*; *Miller et al., 2015*; *Santolini et al., 2000*; *Tebar et al., 1996*; *Figure 2A* and *Figure 2—figure supplement 1C*). However, as proximal to BMP2K-S we also found proteins annotated to ER-Golgi transport, that is SEC16A, SEC24B and ARFGAP1 (*Watson et al., 2006*; *Wendeler et al., 2007*; *Yang et al., 2002*; *Figure 2A* and *Figure 2—figure supplement 1C*). Of note, SEC16A was detected with the highest MS score among all trafficking regulators identified for BMP2K-S (*Figure 2A*).

Unfortunately, we were not able to efficiently immunoprecipitate endogenous SEC16A protein to verify the BioID results, likely due to its large size and lack of appropriate antibodies. Hence, to assess the ability of BMP2K-L or -S to associate with SEC16A, we overexpressed them in HEK293 cells together with EGFP only or with EGFP-tagged SEC16A, and performed immunoprecipitation using anti-GFP antibodies. Although we repeatedly observed some non-specific co-precipitation of both BMP2K isoforms on agarose resin with EGFP only, they were significantly enriched in EGFP-SEC16A precipitates (*Figure 2B*).

Next, we asked whether BMP2K isoforms localized intracellularly to endocytic and/or secretory compartments in K562 cells. Using the monoclonal antibody recognizing in western blotting all BMP2K variants, by confocal microscopy we detected signal predominantly near the PM. There, it overlapped to some extent with EPS15R, found in BioID as proximal to both isoforms (*Figure 2—figure supplement 2A*). However, intensity of the signal detected with anti-BMP2K antibody, although weaker due to BMP2K-L depletion, did not decline upon shBMP2K-S (*Figure 2—figure supplement 2A,B*). Moreover, in cells depleted of all BMP2K variants, the signal reduction, as compared to control cells, was similar to cells lacking BMP2K-L only (*Figure 2—figure supplement 2A,B*). Thus, considering the monoclonal antibody not suitable for detection of BMP2K-S in microscopy, we analyzed the intracellular localization of EGFP-tagged BMP2K variants ectopically expressed in K562 cells. EGFP-BMP2K-L was enriched predominantly near the PM, where it colocalized with EPS15R (*Figure 2C* and *Figure 2—figure supplement 2A,B*). EGFP-BMP2K-S, although also overlapping with EPS15R near the PM, strongly concentrated in the juxtanuclear region positive for SEC16A or SEC24B (*Figure 2C* and *Figure 2—figure supplement 2C*). Such juxtanuclear localization to SEC16A or SEC24B-positive compartment, however weaker, could also be observed for EGFP-BMP2K-L (*Figure 2C* and *Figure 2—figure supplement 2C*).

Altogether, both BMP2K-L and -S can interact with SEC16A and localize to an early secretory compartment, indicating that both variants could regulate SEC16A-dependent intracellular processes.

## BMP2K splicing variants differentially regulate SEC16A protein levels and intracellular distribution

To address whether BMP2K splicing variants could regulate SEC16A-dependent processes, we designed shRNAs to reduce the expression specifically of BMP2K-L or BMP2K-S. Silencing of the

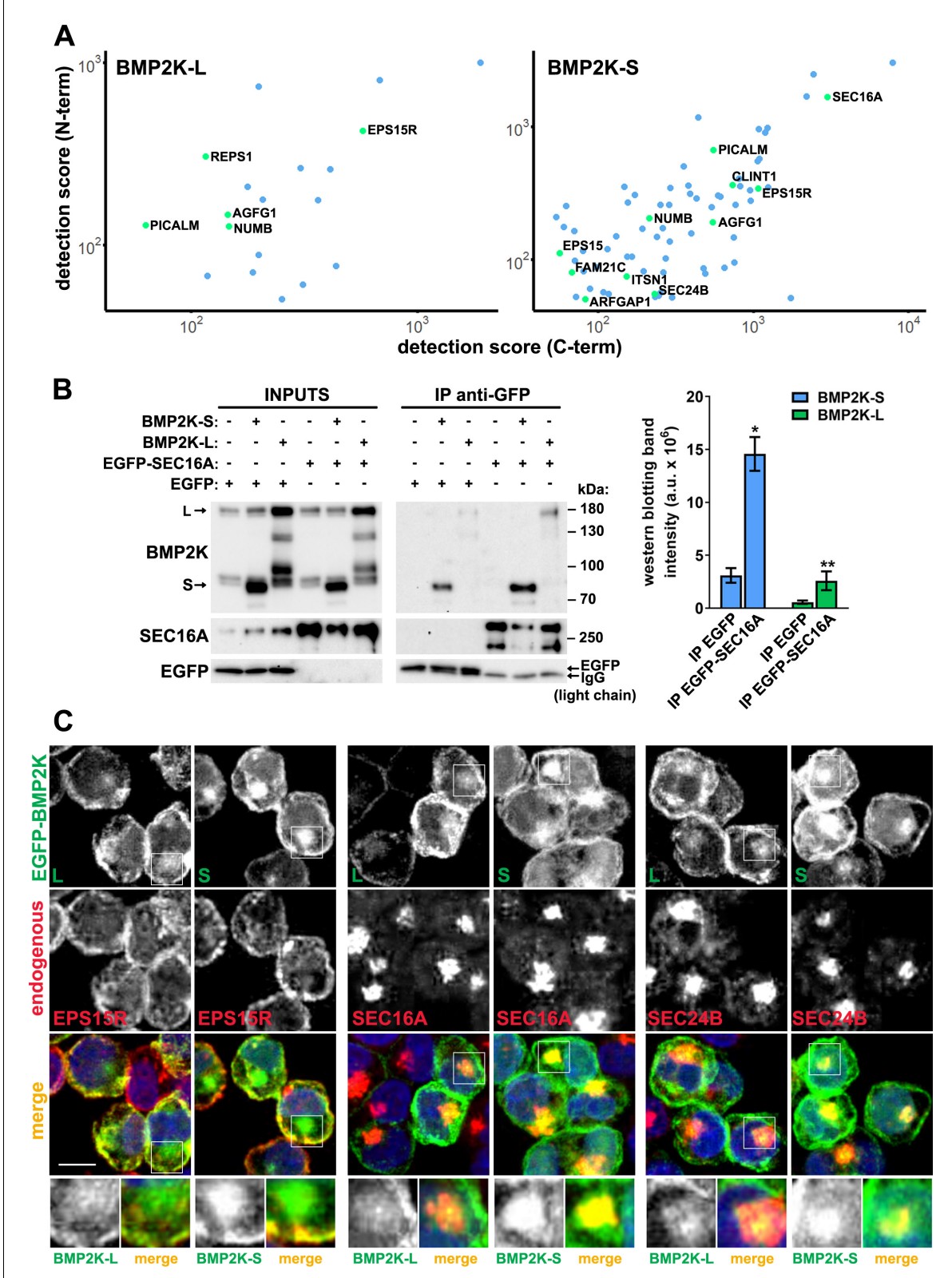

**Figure 2.** The BioID interactome analysis and its subsequent validation show that both BMP2K splicing variants can associate with SEC16A protein and localize to the early secretory compartment. (**A**) Dot plots showing BioID-MS detection scores (log scale) of proteins found as proximal to both N- and C-terminally tagged BMP2K-S or -L variants in HEK293 cells. (**B**) Levels of BMP2K-L, BMP2K-S, EGFP or EGFP-tagged SEC16A in whole cell lysates (INPUTS) or in immunoprecipitates (IP) using anti-GFP antibodies from HEK293 cells. Different combinations of simultaneous ectopic expression of the
*Figure 2 continued on next page*

*Figure 2 continued*

analyzed proteins are indicated above the images. Graph shows non-normalized densitometric analysis of western blotting bands expressed in arbitrary units (a.u.; n = 3 +/- SEM). (C) Maximum intensity projection images from confocal microscope showing localization of ectopically expressed EGFP-tagged BMP2K-L or -S with respect to the indicated proteins and cell nuclei marked with DAPI stain (blue) in K562 cells. Insets: Magnified views of boxed regions in the main images. Scale bars, 10 µm. *p<0.05, **p<0.01.

The online version of this article includes the following source data and figure supplement(s) for figure 2:

**Source data 1.** Numerical data for graphs in *Figure 2*.
**Figure supplement 1.** Proximity biotinylation (BioID) analysis of BMP2K-L or -S interactomes.
**Figure supplement 2.** The analysis of intracellular distribution of BMP2K variants.
**Figure supplement 2—source data 1.** Numerical data for graphs in *Figure 2—figure supplement 2*.

longer variant had no effect on the expression of the shorter, while BMP2K-S depletion only modestly reduced full length BMP2K-L levels (*Figure 3A*).

First, we analyzed whether shRNA-mediated depletion of BMP2K-L or -S affected SEC16A abundance in K562 cells. shBMP2K-L led to a modest but reproducible increase, while shBMP2K-S reduced SEC16A protein levels (*Figure 3A*). Depletion of all BMP2K isoforms (shBMP2K) lowered SEC16A levels, to the same extent as shBMP2K-S (*Figure 3A*). By comparison, protein levels of SEC24B, were essentially unaffected by depletion of BMP2K variants (*Figure 3A*). To ensure that the observed changes in SEC16A abundance were not caused by RNAi off-target effects, we tested additional shRNAs. Reassuringly, shBMP2K-L#2 increased while shBMP2K-S#2 reduced SEC16A amounts (*Figure 3—figure supplement 1A*). The regulation of SEC16A protein levels by the BMP2K-L or -S variants was not due to altered gene transcription. We observed only ~30% downregulation of SEC16A mRNA levels in cells lacking all BMP2K variants (*Figure 3B*).

Next, we tested whether the absence of BMP2K isoforms affected intracellular distribution of SEC16A. In control K562 cells, it concentrated in the juxtanuclear region but was also dispersed throughout the cytoplasm (*Figure 3C*). Depletion of BMP2K-L had no apparent effect on the morphology or staining intensity of juxtanuclear SEC16A compartment but increased integral fluorescence intensity of the dispersed structures, arguing for their expansion (*Figure 3C* and *Figure 3—figure supplement 1B*). Upon shBMP2K-S, SEC16A staining was diffused with lower intensity of the juxtanuclear compartment and higher intensity of the dispersed structures (*Figure 3C* and *Figure 3—figure supplement 1B*). In cells lacking all BMP2K variants, the juxtanuclear SEC16A compartment was also visually diffused but the SEC16A staining intensity was overall lower (*Figure 3C* and *Figure 3—figure supplement 1B*).

Hence, BMP2K-L regulates negatively, while -S positively, SEC16A protein abundance and both variants control SEC16A intracellular distribution. Whether the underlying mechanism involves kinase activities of BMP2K variants or their physical interaction with SEC16A remains to be dissected.

## The BMP2K-L/S system regulates abundance and distribution of COPII assemblies

The observed regulation of SEC16A protein levels and distribution suggested that the two BMP2K variants could affect ERES function and therefore abundance of COPII assemblies. To verify this, we first analyzed the effects of BMP2K variant depletion on distribution of SEC24B, a COPII inner shell component identified as proximal to BMP2K-S in the BioID analysis (*Figure 2A* and *Figure 2—figure supplement 1C*). Similarly to SEC16A, in control cells, SEC24B-positive vesicular structures were concentrated juxtanuclearly or dispersed throughout the cytoplasm (*Figure 3—figure supplement 1C,D*). shBMP2K-L reduced integral intensity of the juxtanuclear but not of the dispersed SEC24B structures (*Figure 3—figure supplement 1C,D*). Upon shBMP2K-S, SEC24B staining was diffused as it was the case for SEC16A, with lower intensity of the juxtanuclear and higher of the dispersed structures (*Figure 3—figure supplement 1C,D*).

As shown in yeast, Sec16 regulates COPII turnover by inhibiting the recruitment of Sec31 outer cage component in a manner modulated by Sec24 (*Bharucha et al., 2013*; *Kung et al., 2012*; *Yorimitsu and Sato, 2012*). Given the altered intracellular distribution of SEC16A and SEC24B upon depletions of BMP2K variants, we tested whether they affected the localization of SEC31A, a ubiquitously expressed COPII marker and Sec31 homologue in mammals (*D'Arcangelo et al., 2013*;

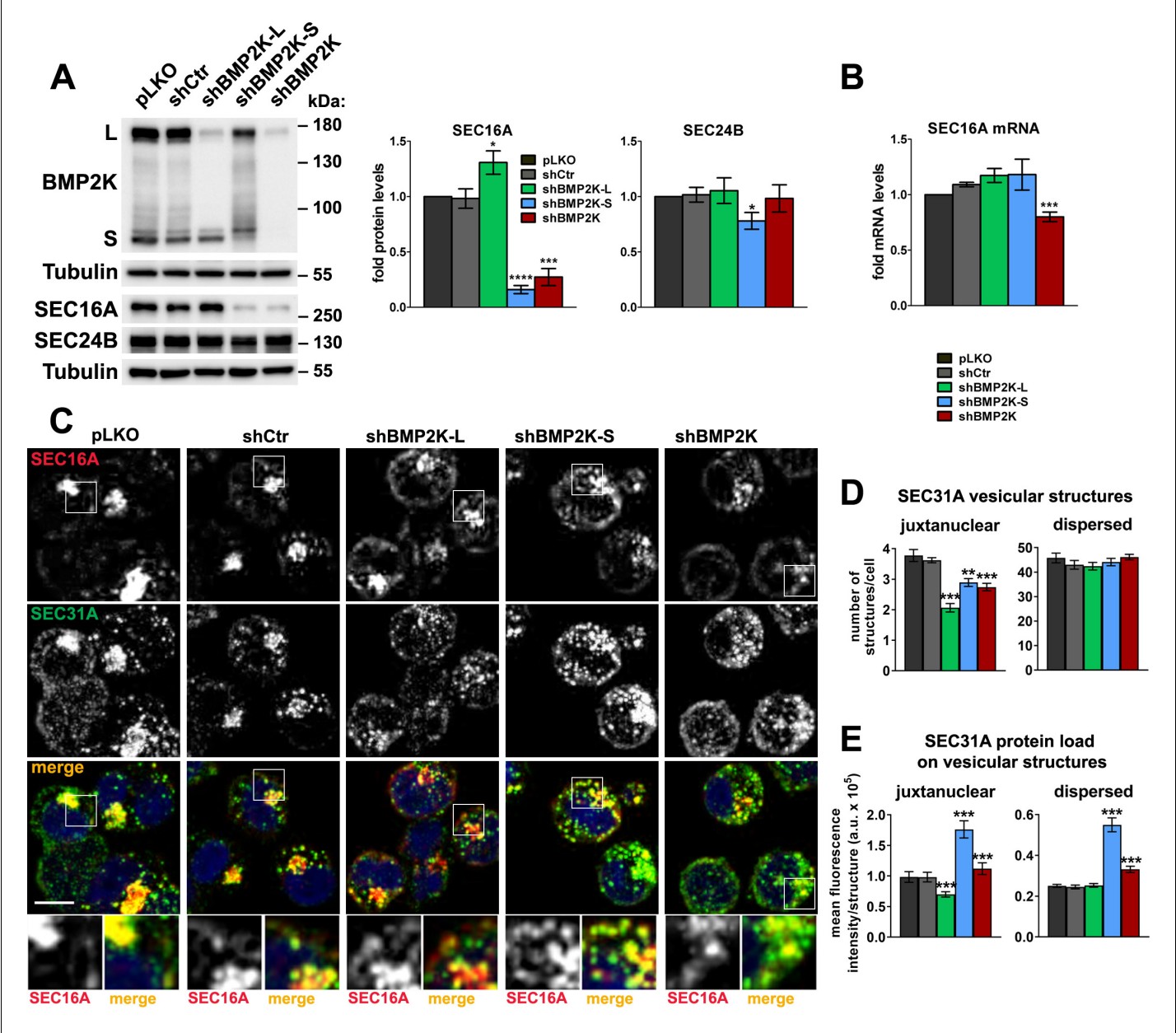

**Figure 3.** BMP2K splicing variants differentially regulate SEC16A protein levels and distribution and control the abundance of SEC31A-positive structures. (**A**) Western blots showing the effect of shRNA-mediated depletion of single (shBMP2K-L or shBMP2K-S) or all BMP2K splicing variants (shBMP2K) on the levels of SEC16A and SEC24B proteins, as compared to empty pLKO vector or non-targeting shRNA construct, shCtr. Graphs show densitometric analysis of western blotting bands for the indicated proteins using tubulin abundance for normalization (n = 5 +/- SEM). (**B**) SEC16A mRNA fold levels in control cells or in cells with shRNA-mediated depletion of BMP2K variants (n = 5 +/- SEM). (**C**) Representative maximum intensity projection images from confocal microscope, showing the effect of shRNA-mediated depletion of BMP2K variants on immunolocalization of SEC16A and SEC31A proteins in K562 cells. Cell nuclei marked with DAPI stain (blue). Insets: Magnified views of boxed regions in the main images. Scale bar, 10 µm. (**D and E**) The number of juxtanuclear and dispersed SEC31A-positive vesicular structures per cell (**D**) or SEC31A mean fluorescence intensity per vesicular structure (SEC31A protein load) presented in arbitrary units (a.u. in E) in control cells or cells lacking BMP2K variants. Quantification from images represented by those in C (n = 5 +/- SEM). *p<0.05, **p<0.01, ***p<0.001, ****p<0.0001.

The online version of this article includes the following source data and figure supplement(s) for figure 3:

**Source data 1.** Numerical data for graphs in *Figure 3*.
**Figure supplement 1.** The effects of depleting BMP2K splicing variants on the abundance and distribution of SEC16A and COPII markers.
**Figure supplement 1—source data 1.** Numerical data for graphs in *Figure 3—figure supplement 1*.

*Satchwell et al., 2013*; *Tang et al., 2000*). In control K562 cells, SEC31A-positive vesicular structures co-localized with both, juxtanuclear and diffused SEC16A- or SEC24B-positive sites (*Figure 3C* and *Figure 3—figure supplement 1C*). shBMP2K-L did not affect the appearance of dispersed SEC31A structures. However, it decreased the number of juxtanuclear structures (*Figure 3D*) and reduced their SEC31A staining intensity (mean fluorescence intensity per structure in *Figure 3E*, hereafter referred to as SEC31A load). shBMP2K-S, although had essentially no effect on the number of SEC31A-positive structures (*Figure 3D*), strongly increased SEC31A load in both, juxtanuclear and dispersed sites (*Figure 3E*).

To confirm whether the observed changes in SEC31A abundance occurred at COPII assemblies, we analyzed whether depletion of BMP2K variants affected specifically SEC31A load at vesicular structures positive for SEC24B (*Figure 3—figure supplement 1C,E*). Reassuringly, upon shBMP2K-L the juxtanuclear SEC24B structures had lower SEC31A load, while upon shBMP2K-S, both juxtanuclear and dispersed SEC24B structures showed elevated SEC31A load (*Figure 3—figure supplement 1C,E*). Hence, we found that BMP2K-L positively regulates the abundance and SEC31A load of juxtanuclear COPII assemblies while BMP2K-S negatively regulates SEC31A load of juxtanuclear and dispersed assemblies.

Having identified these differential roles of the two BMP2K variants, we investigated the intracellular distribution of SEC24B and SEC31A upon shBMP2K that reduced the expression of both variants. shBMP2K recapitulated the effect of shBMP2K-L on SEC24B distribution (lower juxtanuclear SEC24B abundance in *Figure 3—figure supplement 1C,D*) and partially recapitulated the effect of shBMP2K-S on SEC31A abundance (higher SEC31A load in *Figure 3C,E* and *Figure 3—figure supplement 1C,E*). Therefore, cells lacking both BMP2K variants showed lower abundance of juxtanuclear COPII sites, likely due to BMP2K-L depletion, but higher SEC31A load, possibly due to BMP2K-S depletion.

Collectively, we identified a novel intracellular regulatory system, termed the BMP2K-L/S system, where the two BMP2K variants together control the abundance, SEC31A load and distribution of COPII assemblies. Further investigation should address how this dual regulation affects COPII-mediated trafficking.

## BMP2K-L promotes while BMP2K-S restricts autophagic degradation and erythroid differentiation

SEC24B, whose intracellular distribution is regulated by the BMP2K-L/S system (*Figure 3—figure supplement 1C–E*), also contributes to autophagy (*Jeong et al., 2018*). As BMP2K was fished out as a stimulator of LC3-dependent autophagy (*Potts et al., 2013*), we verified whether BMP2K variants regulated autophagic degradation in K562 cells. We observed that shBMP2K-L reduced, while shBMP2K-S increased, the levels of lipidated LC3B (*Figure 4A*), indicative of inhibited or activated autophagy, respectively. shBMP2K upregulated the abundance of LC3B-II, but to a lesser extent than BMP2K-S depletion (*Figure 4A*). This intermediate effect of global BMP2K silencing could result from opposing actions of the two isoforms.

To validate the above results we monitored the levels of SQSTM1/p62 protein (hereafter referred to as p62), an established autophagic cargo. We observed that cells lacking BMP2K-L, although having unchanged p62 mRNA levels (*Figure 4B*), showed a strong increase of its protein content (*Figure 4A*), likely due to restrained autophagy. Conversely, cells lacking BMP2K-S, despite a strong increase in p62 mRNA levels (*Figure 4B*), did not accumulate the protein (*Figure 4A*), which could be explained by elevated autophagic degradation. These observations reinforced the opposing effects of both BMP2K variants on autophagy. In turn, cells treated with shBMP2K showed slightly elevated levels of both p62 protein and mRNA (*Figure 4A,B*), a phenotype difficult to interpret unequivocally.

To clarify the effect of depleting all BMP2K splicing variants on autophagy, we performed autophagic flux analysis. To this end, we inhibited autophagic degradation using bafilomycin A1 (BafA1) and analyzed the accumulation of non-degraded p62 protein in BMP2K-depleted cells (*Figure 4C*). Consistent with autophagy inhibition, in cells lacking BMP2K-L, the accumulation of p62 protein was weaker than in control cells. In turn, in BMP2K-S-depleted cells, stronger p62 accumulation confirmed the increased autophagic flux. Importantly, silencing of all BMP2K splicing variants using shBMP2K also elevated p62 accumulation upon BafA1 treatment, indicative of increased autophagic flux (*Figure 4C*). Hence, we discovered that the two BMP2K isoforms play opposite roles in

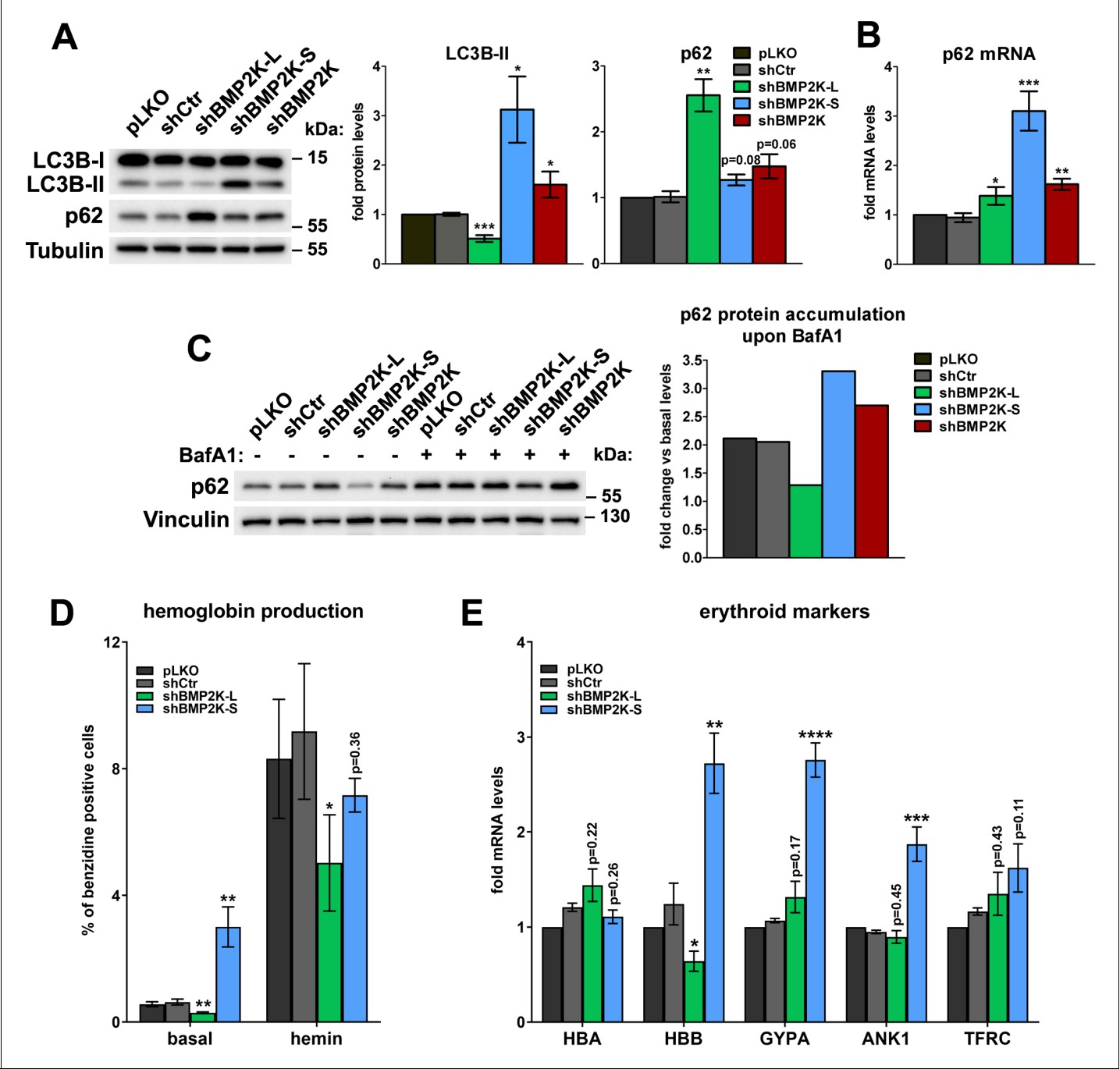

**Figure 4.** BMP2K-L and BMP2K-S differentially regulate autophagic degradation and erythroid differentiation. (**A**) Western blots showing the effects of depletion of single (shBMP2K-L or shBMP2K-S) or all BMP2K splicing BMP2K variants on the levels of LC3B-I, LC3B-II and p62 proteins in K562 cells, as compared to empty pLKO vector or non-targeting shRNA construct, shCtr. Graphs show densitometric analysis of western blotting bands for the indicated proteins using tubulin abundance for normalization (n = 4 +/- SEM). (**B**) p62 mRNA fold change levels in control cells or in cells with shRNA-mediated depletion of BMP2K variants, as in A (n = 5 +/- SEM). (**C**) The abundance of p62 protein in non-treated cells or in cells treated for 15 hr with 75 nM bafilomycin A1 (BafA1). Graph shows calculated fold increase of p62 protein levels induced by BafA1, obtained after densitometric analysis of western blotting bands using vinculin abundance for normalization (n = 1). (**D**) Percentage of benzidine-positive control cells or cells depleted of BMP2K variants, under basal growth conditions or after stimulation for 48 hr with 20 µM hemin (n = 5 +/- SEM). (**E**) Fold changes in mRNA levels of the indicated erythroid markers in control cells or in cells depleted of all BMP2K splicing variants using shRNA (n = 5 +/- SEM). *p<0.05, **p<0.01, ***p<0.001.

The online version of this article includes the following source data and figure supplement(s) for figure 4:

**Source data 1.** Numerical data for graphs in *Figure 4*.

*Figure 4 continued on next page*

*Figure 4 continued*

**Figure supplement 1.** Benzidine staining of K562 cells depleted of BMP2K-L or -S.

regulation of autophagy and that the net outcome of depleting both variants using shBMP2K (resulting in high L/S ratio shown in *Figure 1F*) is modestly elevated basal autophagic degradation.

As autophagy promotes erythroid differentiation (*Cao et al., 2016*; *Grosso et al., 2017*), the effects of BMP2K variant depletion on autophagic degradation suggested that BMP2K-L could favor, while BMP2K-S restrict erythroid differentiation of K562 cells. Consistently, we observed that shBMP2K-L impaired whereas shBMP2K-S increased hemoglobin production under basal culture conditions (*Figure 4D* and *Figure 4—figure supplement 1A*). BMP2K-L-depleted cells also poorly upregulated hemoglobin production upon hemin (*Figure 4D* and *Figure 4—figure supplement 1A*). However, worse differentiation of these cells was not associated with reduced transcription of erythroid-specific genes (*Figure 4E*). In turn, although in cells lacking BMP2K-S, the hemin-induced hemoglobin production was not increased (*Figure 4D* and *Figure 4—figure supplement 1A*), mRNA levels of several erythroid markers were significantly upregulated (*Figure 4E*). Taken together, BMP2K splicing variants differentially regulate erythroid differentiation.

Collectively, we found that BMP2K-L, which positively regulates the abundance of SEC31A at COPII assemblies, stimulates autophagy and erythroid differentiation, while BMP2K-S, which restricts SEC31A presence at COPII assemblies, limits autophagic degradation and erythroid differentiation. It remains to be studied whether the BMP2K-L/-S system controls autophagy via modulation of COPII vesicle trafficking and whether such mechanism underlies regulation of erythroid differentiation by BMP2K variants.

## Discussion

Enrichment of a protein during cell lineage differentiation may reflect its regulatory role in this process, either positive or negative. Induced transcription of *BMP2K* gene was shown in differentiating osteoblasts (*Kearns et al., 2001*) and erythroid cells (*Perucca et al., 2017*), where BMP2K was proposed to inhibit (*Kearns et al., 2001*) or stimulate the differentiation (*Potts et al., 2013*), respectively. Here, we show that BMP2K splicing variants can regulate differentiation in opposite manners. We find that although the protein abundance of BMP2K variants increases at initial steps of erythroid differentiation, it subsequently decreases during red blood cell maturation. As high levels of BMP2K variants restrict erythroid differentiation, the upregulated expression of *BMP2K* gene during erythropoiesis might serve to slow down red blood cell maturation. Consistently, we identify one of the variants, BMP2K-S, as a negative regulator of erythroid differentiation. However, our results indicate that BMP2K-L promotes differentiation. The opposite roles of the two splicing variants provide a possible explanation why erythroid maturation is associated not only with reduction of BMP2K variant levels but also with increased L/S ratio, that is the balance between variant levels shifted in favor of the longer.

As BMP2K kinase has been considered a putative regulator of CME (*Borner et al., 2012*; *Brehme et al., 2009*; *Krieger et al., 2013*), we initially hypothesized that BMP2K variants would control erythroid differentiation via regulation of endocytosis. Yeast homologues of BMP2K and AAK1 kinases, Akl1 and Prk1, suppress endocytosis (*Bar-Yosef et al., 2018*; *Roelants et al., 2017*; *Takahashi et al., 2006*; *Zeng and Cai, 1999*; *Zeng et al., 2001*). Consistently, erythroid differentiation of K562 cells upon silencing of *BMP2K* expression correlates with elevated Tf uptake. However, as measured by the pulse-chase assay, cells lacking BMP2K variants show lower efficiency of Tf endocytosis, possibly due to lower μ2 phosphorylation (*Olusanya et al., 2001*; *Ricotta et al., 2002*; *Wrobel et al., 2019*). Our data suggest that the elevated Tf uptake upon silencing of *BMP2K* expression could result, despite lower endocytosis efficiency, from higher abundance of Tf receptor, whose local clustering promotes Tf endocytosis (*Liu et al., 2010*).

Autophagic clearance of intracellular content is indispensable for erythroid differentiation (*Cao et al., 2016*; *Grosso et al., 2017*) and activation of autophagy can by itself increase expression of genes encoding markers of erythroid maturation (*Cao et al., 2016*). We show that depletion of all BMP2K variants induces autophagic degradation. It also strongly elevates hemoglobin production

upon addition of hemin, a heme precursor which stimulates erythroid differentiation in a manner independent of Tf uptake (*Fibach et al., 1987*) but involving activation of autophagy (*Fader et al., 2016*; *Grosso et al., 2019*). Moreover, a positive regulator of differentiation, BMP2K-L, promotes autophagy, while negative regulator BMP2K-S, inhibits autophagic degradation. Thus, it is tempting to hypothesize that BMP2K variants could regulate erythroid differentiation via modulation of autophagic degradation, that remains to be confirmed.

Although in the BioID interactomes of BMP2K variants we did not find any bona-fide autophagic regulators, we detected components of the ER-Golgi transport pathway that contributes to autophagosome formation (*Ge et al., 2013*; *Shima et al., 2019*; *Wang et al., 2014*). We show that BMP2K-L increases the abundance of SEC24B-positive COPII assemblies and their SEC31A load, while BMP2K-S decreases SEC31A load on SEC24B-positive COPII assemblies and limits their localization to the juxtanuclear secretory compartment. Given the involvement of SEC24B in autophagy (*Jeong et al., 2018*), it is possible that the BMP2K-L/S system controls autophagy in part by regulating trafficking of vesicles containing SEC24B, that remains to be verified. Another intriguing question concerns the cellular role of COPII assemblies dispersed outside of the juxtanuclear secretory compartment, whose SEC31A load strongly increases in the absence of BMP2K-S. Once COPII vesicles are formed, they immediately reach their target compartment, ERGIC, located in close proximity to ERES (*Lord et al., 2013*). Therefore it is unlikely that the dispersed COPII assemblies represent vesicles transported from the juxtanuclear compartment to the cell periphery. Future studies should address whether they are implicated in regulation of autophagic degradation or erythroid differentiation by BMP2K variants.

Although K562 cells represent a convenient model of erythroid differentiation, they do not fully recapitulate distinct maturation stages occurring physiologically. However, our finding that depletion of both BMP2K variants in these cells promotes erythroid maturation is consistent with reduction of BMP2K-L and -S levels upon late stages of mouse fetal liver cell differentiation. Erythroblast maturation involves activation of autophagic degradation (*Zhang et al., 2009*) and secretion of erythroid-specific markers to the PM (*Satchwell et al., 2013*; *van den Akker et al., 2010*). Hence, our results suggest that the BMP2K-L/S system could act in coordination of these events during maturation of the erythroid lineage.

Endocytosis and autophagy (*Fraser et al., 2017*; *Tooze et al., 2014*) as well as secretory trafficking and autophagy (*Davis et al., 2017*; *McCaughey and Stephens, 2018*) are clearly interdependent. We provide evidence that an endocytic kinase regulates autophagy. This may occur at least in part via modulation of COPII assembly. Thus, through its alternative splicing variants, BMP2K kinase functions at the crossroad between endocytosis, secretion and autophagy. We propose that BMP2K variants represent a regulatory system wherein the activator (BMP2K-L) promotes, while the inhibitor (BMP2K-S) restricts processes required for erythroid maturation.

# Materials and methods

**Key resources table**

| Reagent type (species) or resource | Designation | Source or reference | Identifiers | Additional information |
|---|---|---|---|---|
| Cell line (*Homo sapiens*) | K562 | ATCC | Cat# CCL-243, RRID:CVCL_0004 | |
| Cell line (*Homo sapiens*) | HEK293 | ATCC | Cat# CRL-1573, RRID:CVCL_0045 | |
| Cell line (*Homo sapiens*) | HEK293T | ATCC | Cat# CRL-3216, RRID:CVCL_0063 | |
| Transfected construct (*Homo sapiens*) | BMP2K shRNA | Sigma-Aldrich | TRCN0000000915 | pLKO.1 Lentiviral construct to transfect and express the shRNA |

*Continued on next page*

*Continued*

| Reagent type (species) or resource | Designation | Source or reference | Identifiers | Additional information |
|---|---|---|---|---|
| Transfected construct (*Homo sapiens*) | BMP2K-L shRNA | This paper | | pLKO.1 Lentiviral construct to transfect and express the shRNA. See *Supplementary file 1*-Table 3. |
| Transfected construct (*Homo sapiens*) | BMP2K-S shRNA | This paper | | pLKO.1 Lentiviral construct to transfect and express the shRNA. See *Supplementary file 1*-Table 3. |
| Transfected construct (*Homo sapiens*) | empty pLKO.1 plasmid | Sigma-Aldrich | SHC001 | pLKO.1 Lentiviral construct to transfect and express the shRNA |
| Transfected construct (*Homo sapiens*) | non-targeting shRNA plasmid | Sigma-Aldrich | SHC202 | pLKO.1 Lentiviral construct to transfect and express the shRNA |
| Transfected construct (*Homo sapiens*) | BMP2K gRNAs (gBMP2K#1and #2) | This paper | | lentiCRISPRv2 Lentiviral construct to transfect and express the gRNA together with Cas9. See *Supplementary file 1*-Table 4. |
| Transfected construct (*Homo sapiens*) | non-targeting gRNAs (gCtr#1and #2) | This paper | | lentiCRISPRv2 Lentiviral construct to transfect and express the gRNA together with Cas9. See *Supplementary file 1*-Table 4. |
| Transfected construct (*Homo sapiens*) | pEGFP-SEC16A | Addgene | Cat# 36155 | |
| Biological sample (*Mus musculus*) | Primary mouse fetal liver cells | Mossakowski Medical Research Centre Polish Academy of Sciences | | Freshly isolated from *Mus musculus* (strain C57BL/6J) |
| Antibody | anti-BMP2K (mouse monoclonal, ascites) | Santa Cruz Biotechnology | Cat# sc-134284, RRID:AB_2227882 | WB(1:2000), IF(1:500) |
| Antibody | anti-SEC16A (rabbit polyclonal) | Bethyl | Cat# A300-648A, RRID:AB_519338 | WB(1:1000) |
| Antibody | anti-SEC16A (rabbit polyclonal) | Atlas Antibodies | Cat# HPA005684, RRID:AB_1079189 | IF(1:400) |
| Antibody | anti-SEC24B (rabbit monoclonal) | Cell Signaling Technology | Cat# 12042, RRID:AB_2797807 | WB(1:1000), IF(1:200) |
| Antibody | anti-SEC31A (mouse monoclonal) | BD Biosciences | Cat# 612350, RRID:AB_399716 | IF(1:200) |
| Antibody | anti-EPS15R (rabbit monoclonal) | Abcam | Cat# ab76004, RRID:AB_1310187 | IF(1:200) |
| Antibody | anti-LC3B (rabbit polyclonal) | Cell Signaling Technology | Cat# 2775, RRID:AB_915950 | WB(1:1000) |
| Antibody | anti-SQSTM1/p62 (mouse monoclonal) | BD Biosciences | Cat# 610833, RRID:AB_398152 | WB(1:1000) |
| Antibody | anti-Phospho-$\mu2$ (Thr156) (D4F3) (rabbit monoclonal) | Cell Signaling Technology | Cat# 7399, RRID:AB_10949770 | WB(1:1000) |
| Antibody | anti-$\mu2$ (mouse monoclonal) | BD Biosciences | Cat# 611350, RRID:AB_398872 | WB(1:500) |
| Antibody | anti-GFP (goat polyclonal) | R&D Systems | Cat# AF4240, RRID:AB_884445 | IP(1:60) |

*Continued*

| Reagent type (species) or resource | Designation | Source or reference | Identifiers | Additional information |
|---|---|---|---|---|
| Antibody | APC-conjugated anti-CD71/TFRC (rat monoclonal) | Thermo Fisher Scientific | Cat# 17-0711-80, RRID:AB_1834356 | FC(1:160) |
| Antibody | PE-conjugatedanti-TER-119 (rat monoclonal) | Thermo Fisher Scientific | Cat# 12-5921-81, RRID:AB_466041 | FC(1:80) |
| Peptide, recombinant protein | recombinant human EPO | PeproTech | Cat# 100–64 | |
| Chemical compound, drug | Hemin | Sigma-Aldrich | Cat# H9039 | |
| Chemical compound, drug | benzidine dihydrochloride | Sigma-Aldrich | Cat# B3383 | |
| Chemical compound, drug | bafilomycin A1 | Sigma-Aldrich | Cat# B1793 | |
| Software, algorithm | Harmony 4.9 | PerkinElmer | | Imaging and Analysis Software for Opera Phenix microscope |
| Software, algorithm | ImageJ software | ImageJ (http://imagej.nih.gov/ij/) | RRID:SCR_003070 | |
| Software, algorithm | GraphPad Prism eight software | GraphPad Prism (https://graphpad.com) | RRID:SCR_015807 | |
| Other | Alexa Fluor 647-conjugated Transferrin | Thermo Fisher Scientific | Cat# T23366 | |
| Other | Alexa Fluor 488-conjugated 10 kDa dextran | Thermo Fisher Scientific | Cat# T23366 | |
| Other | Dynabeads MyOne Streptavidin-coupled magnetic beads | Thermo Fisher Scientific | Cat# 65001 | |

## Antibodies

The following antibodies were used: anti-BMP2K (sc-134284) and anti-GAPDH (sc-25778) from Santa Cruz; anti-phospho-μ2 (#7399), anti-SEC24B (#12042) and anti-LC3B (#2775) from Cell Signaling Technologies; anti-EPS15R (ab76004) from Abcam; anti-tubulin (T5168), anti-vinculin (V9131) and anti-beta-actin (A5441) from Sigma; anti-SQSTM1/p62 (610833), anti-μ2 (611350) and anti-SEC31A (612350) from BD Biosciences; anti-GFP (AF4240) from R&D Systems; anti-SEC16A (A300-648A) from Bethyl Laboratories Inc; anti-SEC16A (HPA005684) from Atlas Antibodies; anti-TFRC (H68.4), APC-conjugated anti-TFRC (#17-0711-80) and PE-conjugated anti-Ter-119 (#12-5921-81) from Thermo Fisher Scientific; secondary horseradish peroxidase (HRP)-conjugated goat anti-mouse and goat anti-rabbit antibodies from Jackson ImmunoResearch; secondary Alexa Fluor 488-conjugated anti-mouse and Alexa Fluor 555-conjugated anti-rabbit antibodies from Thermo Fisher Scientific.

## Plasmids

To obtain pcDNA3.1-BMP2K-L construct, full-length human BMP2K-L was amplified from HEK293 cell cDNA by PCR using oligonucleotides 5'-ggggAAGCTTATGAAGAAGTTCTCTCGGATGCC-3' (forward with HindIII restriction site) and 5'-ggggGGATCCCTACTGTTTAGAAGGAAATGGAGCAG-3' (reverse with BamHI restriction site), subcloned into the pcDNA3.1 vector, and sequence-verified. To obtain pcDNA3.1-BMP2K-S construct, full-length human BMP2K-S was amplified from the clone IRAT32H09 (SourceBioScience) by PCR with the oligonucleotides 5'-ggggAAGCTTATGAAGAAG TTCTCTCGGATGCC-3' (forward with HindIII restriction site) and 5'-AACAGCTATGACCATG-3' (reverse M13 primer), subcloned into the pcDNA3.1 vector, and sequence-verified.

The pcDNA3.1-Myc-BirA*-BMP2K-L or pcDNA3.1-Myc-BirA*-BMP2K-S constructs were obtained by amplification of BMP2K-L or BMP2K-S from pcDNA3.1-BMP2K-L or pcDNA3.1-BMP2K-S using

oligonucleotides 5'-gctaGGATCCTATGAAGAAGTTCTCTCGGAT-3' (forward with BamHI restriction site) and 5'-gcgcGGTACCCTACTGTTTAGAAGGAAATG-3' (reverse with KpnI restriction site for BMP2K-L) or 5'-gcgcGGTACCTTACTGTGAAGCAAAATAAG-3' (reverse with KpnI restriction site for BMP2K-S) and subcloning into pcDNA3.1 mycBioID vector. The pcDNA3.1-BMP2K-L-BirA*-HA or pcDNA3.1-BMP2K-S-BirA*-HA constructs were obtained by amplification of BMP2K-L or BMP2K-S from pcDNA3.1-BMP2K-L or pcDNA3.1-BMP2K-S using oligonucleotides 5'-gcgcACCGGTATGAA-GAAGTTCTCTCGGAT-3' (forward with AgeI restriction site) and 5'-gcgcGGATCCCTGTTTAGAAG-GAAATGGAG-3' (reverse with BamHI restriction site for BMP2K-L) or 5'-gcgcGGATCCCTG TGAAGCAAAATAAGCCT-3' (reverse with BamHI restriction site for BMP2K-S) and subcloning into pcDNA3.1 MCS-BirA(R118G)-HA vector. The pcDNA3.1 mycBioID and pcDNA3.1 MCS-BirA (R118G)-HA vectors (Addgene plasmids # 35700 and # 36047) were gifts from Kyle Roux (*Satchwell et al., 2011*). pEGFP-SEC16A construct (Addgene plasmid # 36155) was a gift from David Stephens (*Watson et al., 2006*). psPAX2 (Addgene plasmid # 12260) and pMD2.G (Addgene plasmid # 12259) lentiviral packaging plasmids were a gift from Didier Trono. pUltra-Chili vector (Addgene plasmid # 48687) was a gift from Malcolm Moore. MISSION shRNA plasmids were obtained from Sigma-Aldrich. pLKO.1 - TRC cloning vector (Addgene plasmid # 10878) was a gift from David Root. LentiCRISPRv2 vector (Addgene plasmid # 52961) was a gift from Feng Zhang. shRNA sequences for depletion of specific BMP2K splicing variants were cloned into the pLKO.1 - TRC cloning vector using a protocol provided by Addgene. gRNA sequences for CRISPR/Cas9 mediated gene inactivation were cloned into the LentiCRISPRv2 vector using a protocol described elsewhere (*Sanjana et al., 2014*).

To obtain the pUltra-EGFP-BMP2K-L or pUltra-EGFP-BMP2K-S lentiviral constructs, first, pEGFP-BMP2K-L and pEGFP-BMP2K-S plasmids were generated. The pEGFP-BMP2K-L construct was obtained by restriction digestion of pcDNA3.1-BMP2K-L with HindIII and BamHI and subcloning the insert into pEGFP-C3 vector. The pEGFP-BMP2K-S construct was obtained by amplification of BMP2K-S from pcDNA3.1-BMP2K-S using oligonucleotides 5'-ggggAAGCTTATGAAGAAGTTCTC TCGGATGCC-3' (forward with HindIII restriction site) and 5'-ggggGGATCCTTACTGTGAAGCAAAA TAAGCCTTC-3' (reverse with BamHI restriction site) and subcloning into pEGFP-C3 vector. Next, pEGFP-BMP2K-L or pEGFP-BMP2K-S plasmids were digested with AgeI and BamHI restriction enzymes and the inserts were subcloned into pUltra-Chili vectors, substituting their dTomato inserts with the EGFP-BMP2K coding sequences.

## Cell culture and treatment

HEK293 cells were maintained in Dulbecco's modified Eagle's medium (DMEM) and K562 cells in RPMI-1640 medium (Sigma-Aldrich). All media had high glucose concentration and were supplemented with 10% fetal bovine serum and 2 mM L-glutamine (Sigma-Aldrich). Hemin (H9039, Sigma-Aldrich) was used to stimulate erythroid differentiation of K562 cells at 20 µM concentration for 48 hr or 72 hr. Hemin 4 mM stocks were prepared according to the published protocol (*Addya et al., 2004*). Briefly, 13 mg of hemin was resuspended in 200 µl of 0.5 M sodium hydroxide, mixed with 250 µl of 1 M Tris (pH 7.8) and $H_2O$ was added to a final volume of 5 ml. K562 cells were also treated for 48 hr with 75 nM bafilomycin A1 (B1793, Sigma-Aldrich) to assess autophagic flux.

The identity of K562 cells has been authenticated by STR profiling. Their profile matched all of the reference ATCC STR loci. These cells were regularly tested as mycoplasma-negative.

## Cell transfection and lentiviral transduction

For western blotting in HEK293 cells, $2.6*10^5$ cells were seeded per well in 12-well plates. For co-immunoprecipitation of proteins ectopically expressed in HEK293 cells, $6*10^5$ cells were seeded in 60 mm dishes. The cells were transfected after 24 hr with plasmid DNA using Lipofectamine 2000 Transfection Reagent (Thermo Fisher Scientific) according to the manufacturer's protocol.

Lentiviral particles to transduce K562 cells were produced in HEK293T cells using psPAX2 and pMD2.G packaging plasmids as described elsewhere (*Barde et al., 2010*). For infection, $1*10^6$ K562 cells were grown in 10 ml of virus-containing RPMI-1460 medium for 24 hr. To achieve shRNA-mediated depletion of BMP2K, MISSION shRNA plasmids were used. The empty pLKO.1 plasmid (SHC001), and the construct expressing non-targeting shRNA (SHC202) served as controls. After initial testing (not shown in the manuscript figures) of 5 different shRNA sequences (TRCN0000000914,

TRCN0000000915, TRCN0000000916, TRCN0000000917, TRCN0000226438), only one (TRCN0000000915, shBMP2K) was found to efficiently downregulate BMP2K expression as assessed by western blotting. To achieve shRNA-mediated depletion of specific BMP2K splicing variants, shRNA sequences were designed using the siRNA Selection Program (*Yuan et al., 2004*) and cloned into pLKO.1 – TRC cloning vector. These sequences are listed in *Supplementary file 1*-Table 3. To achieve CRISPR/Cas9-mediated inactivation of *BMP2K* gene, four different gRNA sequences (*Doench et al., 2016*) were cloned into the lentiCRISPRv2 vector and were lentivirally introduced into K562 cells. Their efficiency of gene expression silencing was tested by western blotting and two sequences causing the strongest reduction of BMP2K protein levels were chosen for further experiments. gRNAs used in this study are listed in *Supplementary file 1*-Table 4. For BMP2K depletion, cells were transduced with lentiviral particles containing control or gene-targeting vectors for 24 hr and selected with 2 µg/ml puromycin for 72 hr.

For overexpression of EGFP-tagged BMP2K variants, K562 cells were transduced for 24 hr with lentiviral particles containing pUltra-EGFP-BMP2K-L or -S plasmids and analyzed after three subsequent days of culture.

## Mouse fetal liver erythroblast isolation, ex vivo differentiation and FACS

Mouse fetal liver erythroid progenitors were isolated and stimulated for differentiation as described (*Zhao et al., 2014*). Briefly, 8–10 livers were extracted from embryonic day 13.5 C56/BL6 mouse embryos and manually disrupted. Ter119-positive differentiated cells were recognized using specific biotin-conjugated antibodies (13-5921-81) and removed from the cell suspension using Dynabeads MyOne Streptavidin-coupled magnetic beads (65001; both reagents from Thermo Fisher Scientific). To induce erythroid differentiation, $4*10^5$ purified and washed cells were cultured on fibronectin-coated 12-well plates in the presence of erythropoietin (EPO)-containing Iscove's Modified Dulbecco's Medium: 2 U/ml recombinant human EPO (100-64), 10 µg/ml insulin (I3536), 200 µg/ml holo-transferrin (T0665), 0.1 mM β-mercaptoethanol (M3148), 1% BSA, 15% FBS and 2 mM L-glutamine (EPO from PeproTech and other reagents from Sigma-Aldrich). After 24 hr, 48 hr or 72 hr of differentiation the cells were harvested for western blotting or qRT-PCR analyses.

To analyze protein levels in mouse erythroblasts at various stages of differentiation, the isolated mouse fetal liver progenitors were differentiated for 96 hr and labeled with APC-conjugated anti-TFRC and PE-conjugated anti-Ter-119 antibodies as described (*Zhao et al., 2014*). The labeled cells were FACS-separated using BD FACSAria II cell sorter and harvested for western blotting analysis.

## Immunofluorescence staining and microscopy

K562 cells (non-treated or transduced with lentiviral plasmids) were transferred to ice, fixed with 3% paraformaldehyde for 15 min on ice followed by 15 min at room temperature. After three washes with PBS, cells were immunostained in suspension using an adapted version of a protocol described for adherent cells (*Mamińska et al., 2016*). Stained cells were resuspended in 0.5% low-melting agarose (Sigma-Aldrich) and transferred to microscopy 96-well plates (Greiner Bio-One). The plates were scanned using Opera Phenix high content screening microscope (PerkinElmer) with 40 × 1.1 NA water immersion objective. Harmony 4.9 software (PerkinElmer) was used for image acquisition and their quantitative analysis. To quantify chosen parameters in the obtained images (number of vesicular structures per cell, mean fluorescence intensity per structure, integral fluorescence intensity per cell and mean pixel intensity), more than 40 microscopic fields were analyzed per each experimental condition. Maximum intensity projection images were obtained from 7 to 8 z-stack planes with 1 µm interval. Pictures were assembled in Photoshop (Adobe) with only linear adjustments of contrast and brightness.

## Western blotting

Cells were lysed in RIPA buffer (1% IGEPAL CA-630, 0.5% sodium deoxycholate, 0.1% SDS, 50 mM Tris (pH 7.4), 150 mM NaCl, 0.5 mM EDTA) supplemented with protease inhibitor cocktail (6 µg/ml chymostatin, 0.5 µg/ml leupeptin, 10 µg/ml antipain, 2 µg/ml aprotinin, 0.7 µg/ml pepstatin A and 10 µg/ml 4-amidinophenylmethanesulfonyl fluoride hydrochloride; Sigma-Aldrich) and phosphatase inhibitor cocktails (P0044 and P5726, Sigma-Aldrich). For HEK293 or K562 cells, protein

concentration was measured with BCA Protein Assay Kit (Thermo Scientific) and 10–50 µg of total protein per sample were resolved on SDS-PAGE. For mouse fetal liver erythroblasts, RIPA lysate equivalents of $3*10^5$ cells were resolved on SDS-PAGE. Resolved proteins were transferred to nitro-cellulose membrane (Whatman), probed with specific primary and secondary antibodies, and detected using ChemiDoc imaging system (Bio-Rad) or Odyssey infrared imaging system (LI-COR Biosciences).

Densitometric analysis of western blotting bands was performed using Image Lab 5.2.1 software (Bio-Rad). The raw data were normalized to tubulin band intensities (for K562 cells) or were not normalized (for mouse fetal liver erythroblasts) and presented as fold levels over respective controls.

## Co-immunoprecipitation (Co-IP)

For co-immunoprecipitation of proteins ectopically expressed in HEK293 cells, an equivalent of $8*10^5$ cells in 300 µl of RIPA buffer was used per reaction. The lysates were incubated for 1.5 hr at 4°C with 1 µg of goat anti-GFP antibodies with rotation. The immune complexes were captured by incubation with Protein G-agarose for 2 hr at 4°C with rotation. The agarose beads-bound protein complexes were spun down and washed three times using IP wash buffer (50 mM Hepes, pH 7.5, 300 mM NaCl, 1 mM EGTA, 1 mM EDTA, 1% Triton X-100, 10% glycerol, 5 µg/ml DNase and protease inhibitor cocktail) and one time using 50 mM Hepes, pH 7.5. The proteins were eluted from the beads by incubation at 95°C for 10 min with Laemmli buffer and analyzed by western blotting. Protein G-agarose beads were blocked in PBS with 5% BSA for 2 hr at 4°C with rotation before they were added to capture the immune complexes.

## Quantitative real-time PCR (qRT-PCR)

Total RNA was isolated from cells with High Pure Isolation Kit (Roche). For cDNA synthesis random nonamers, oligo(dT)23 and M-MLV reverse transcriptase (Sigma-Aldrich) were used according to the manufacturer's instructions. To estimate the expression of genes of interest we used primers designed with NCBI tool (and custom-synthesized by Sigma-Aldrich) listed in *Supplementary file 1*-Table 5. The qRT-PCR reaction was performed with the Kapa Sybr Fast ABI Prism qPCR Kit (KapaBiosystems) using a 7900HT Fast Real-Time PCR thermocycler (Applied Biosystems) with at least three technical repeats per experimental condition. The data were normalized according to the expression level of housekeeping gene, *GAPDH* (in K562 cells) or *RPL19* (in mouse fetal liver erythroblasts) and presented as fold changes.

## Transferrin and dextran uptake assay

For continuous transferrin (Tf) uptake experiments, $1*10^6$ K562 cells were pre-incubated for 30 min on ice with 25 µg/ml Alexa Fluor 647-conjugated Tf (Thermo Fisher Scientific) and incubated at 37 °C for 5 or 40 min (early uptake or steady-state loading, respectively) followed by washing and fixation with 3% PFA for 5 min. For dextran uptake measurement, cells were incubated with 125 µg/ml Alexa Fluor 488-conjugated 10 kDa dextran (Thermo Fisher Scientific) at 37 °C for 40 min followed by washing and fixation.

The pulse-chase Tf uptake was performed according to a described protocol (*Dannhauser et al., 2017*) but adapted for K562 cells cultured in suspension. Briefly, non-starved cells were incubated for 15 min on ice with 5 µg/ml Alexa Fluor 647-conjugated Tf (Thermo Fisher Scientific) in RPMI medium, followed by washing unbound Tf with ice-cold PBS. Then cells were either analyzed immediately by flow cytometry, to assess the cell surface Tf binding, or were resuspended in pre-warmed RPMI medium and incubated at 37 °C for 5 min to initiate Tf uptake. The uptake was stopped by washing in ice-cold PBS and non-internalized Tf was stripped by two acid washes (0.15 m glycine buffer, pH 3). Upon final wash with ice-cold PBS, cells were analyzed by flow cytometry. For each experimental conditions we applied control conditions, including non-stained cells (without Alexa Fluor 647-conjugated Tf incubation to estimate background fluorescence), and cells stripped immediately upon incubation with Tf (to determine the efficiency of surface-bound Tf removal).

Fixed (continuous uptake) or non-fixed (pulse-chase uptake) cells were resuspended in PBS and the fluorescence of incorporated Tf or dextran was recorded from 50 000 cells with a BD LSR Fortessa flow cytometer. The data collection and calculation of mean fluorescence intensities were

performed using BD FACSDiva 6.2 software. Flow cytometry data were analyzed and visualized by FlowJo software (BD Biosciences).

## Hemoglobin content analysis

Hemoglobin content in K562 cells was visualized by staining with benzidine dihydrochloride (B3383, Sigma-Aldrich). Briefly, 300 µl of 2 mg/ml solution in 3% acetic acid was added to an equal volume of fresh RPMI-1640 medium with $2*10^5$ cells, followed immediately by addition of 12 µl of 30% hydrogen peroxide solution. The stained cells were imaged in bright-field using the IX 7C Olympus microscope and the percentage of hemoglobin-positive cells was counted using the ImageJ software. At least 1000 cells from 4 to 6 images were counted per sample.

## Proximity biotinylation (BioID)

The expression of Myc-BirA*-BMP2K-L, Myc-BirA*-BMP2K-S, BMP2K-L-BirA*-HA or BMP2K-S-BirA*-HA fusion proteins was assessed after transient transfection of HEK293 cells. The obtained constructs as well as empty vectors were linearized using PvuI restriction enzyme and introduced into HEK293 cells using Lipofectamine 2000 Transfection Reagent. The cells underwent antibiotic selection with 500 µg/ml G418 (Gibco) and clones stably expressing each of the fusion proteins were generated. The expression efficiencies among the obtained clones were assessed by western blotting (not shown in the manuscript figures) and two clones expressing the highest levels of each fusion protein were selected for biotin labelling. Biotin labelling and subsequent pull down of biotinylated proteins using Dynabeads MyOne Streptavidin-coupled magnetic beads (65001; Thermo Fisher Scientific) was performed in two biological repeats for each of the two selected clones, following the originally described protocol (*Roux et al., 2013*). The qualitative analysis of biotinylated proteins from each sample was performed by mass spectrometry and analyzed using Mascot software. The Mascot scores obtained for each of the two clones per condition (Myc-BirA*-BMP2K-L, Myc-BirA*-BMP2K-S, BMP2K-L-BirA*-HA or BMP2K-S-BirA*-HA) were averaged and the scores obtained for proteins identified in the control samples (false-positive background of Myc-BirA* or BirA*-HA) were subtracted. The final subtracted C-tag or N-tag scores were obtained from averaging the subtracted scores from two biological repeats.

## Mass spectrometry of biotinylated proteins

Proteins biotinylated in BioID were reduced with 5 mM TCEP (for 60 min at 60°C). To block reduced cysteines, 200 mM MMTS at a final concentration of 10 mM was added and the sample was incubated at room temperature for 10 min. Trypsin (Promega) was added at a 1:20 vol./vol. ratio and incubated at 37 °C overnight. Finally, trifluoroacetic acid was used to inactivate trypsin. Peptide mixtures were analyzed by liquid chromatography coupled to tandem mass spectrometry (LC-MS/MS) using Nano-Acquity (Waters Corporation) UPLC system and LTQ-FT-Orbitrap (Thermo Scientific) mass spectrometer. Measurements were carried out in the positive polarity mode, with capillary voltage set to 2.5 kV. A sample was first applied to the Nano-ACQUITY UPLC Trapping Column using water containing 0.1% formic acid as a mobile phase. Next, the peptide mixture was transferred to Nano-ACQUITY UPLC BEH C18 Column using an acetonitrile gradient (5–35% acetonitrile over 160 min) in the presence of 0.1% formic acid with a flow rate of 250 nl/min. Peptides were eluted directly to the ion source of the mass spectrometer. Each LC run was preceded by a blank run to ensure that there was no carry-over of material from previous analysis. HCD fragmentation was used. Up to 10 MS/MS events were allowed per each MS scan.

Acquired raw data were processed by Mascot Distiller followed by Mascot Search (Matrix Science, London, UK, on-site license) against SwissProt database restricted to human sequences. Search parameters for precursor and product ions mass tolerance were 30 ppm and 0.1 Da, respectively, enzyme specificity: trypsin, missed cleavage sites allowed: 1, fixed modification of cysteine by methylthio and variable modification of methionine oxidation. Peptides with Mascot score exceeding the threshold value corresponding to <5% expectation value, calculated by Mascot procedure, were considered to be positively identified. BioID-MS detection scores were visualized using R package ggplot2 and scales (R version 3.4.4).

The mass spectrometry proteomics data have been deposited to the ProteomeXchange Consortium via the PRIDE (*Perez-Riverol et al., 2019*) partner repository with the dataset identifier PXD013542.

## Statistical analysis

Data are provided as mean ± SEM. Statistical analysis was performed with Prism 8 (GraphPad Software) using unpaired two-tailed Student t test (for qRT-PCR analysis and western blotting densitometry in K562 cells) or paired two-tailed Student t test (for uptake experiments, hemoglobin content analysis, western blotting densitometry in mouse fetal erythroblasts and quantified parameters from confocal microcopy analysis). Data points were marked according to the p value, where $p > 0.05$ is left unmarked or indicated with p=value, $*p < 0.05$, $**p < 0.01$, $***p < 0.001$, $****<0.0001$.

## Acknowledgements

We are grateful to J Jaworski and K Mleczko-Sanecka for providing reagents. We also thank M Banach-Orłowska, M Maksymowicz, A Poświata, L Wolińska-Nizioł and D Zdżalik-Bielecka for critical reading of the manuscript. This work was funded by the MAESTRO grant (UMO-2011/02/A/NZ3/00149) from National Science Center to M Miaczynska. M Miaczynska, M Kaczmarek and K Jastrzębski were supported by TEAM grant (POIR.04.04.00-00-20CE/16–00), J Cendrowski and M Mazur supported by HOMING grant (POIR.04.04.00-00-1C54/16-00), K Piwocka was supported by TEAM-TECH Core Facility Plus grant (POIR.04.04.00-00-23C2/17-00) – the three grants from the Foundation for Polish Science co-financed by the European Union under the European Regional Development Fund. Mass spectrometric analysis was performed in the Mass Spectrometry Laboratory, IBB PAS, Warsaw. The MS equipment was sponsored in part by the Centre for Preclinical Research and Technology (CePT), a project co-sponsored by European Regional Development Fund and Innovative Economy, The National Cohesion Strategy of Poland.

## Additional information

### Funding

| Funder | Grant reference number | Author |
| --- | --- | --- |
| Narodowe Centrum Nauki | UMO-2011/02/A/NZ3/00149 | Jaroslaw Cendrowski<br>Katarzyna Kuzmicz-Kowalska<br>Kamil Jastrzebski<br>Marta Miaczynska |
| Fundacja na rzecz Nauki Polskiej | POIR.04.04.00-00-20CE/16-00 | Marta Kaczmarek<br>Kamil Jastrzebski<br>Marta Miaczynska |
| Fundacja na rzecz Nauki Polskiej | POIR.04.04.00-00-1C54/16-00 | Jaroslaw Cendrowski<br>Michał Mazur |
| Fundacja na rzecz Nauki Polskiej | POIR.04.04.00-00-23C2/17-00 | Katarzyna Piwocka |

The funders had no role in study design, data collection and interpretation, or the decision to submit the work for publication.

### Author contributions

Jaroslaw Cendrowski, Conceptualization, Data curation, Formal analysis, Supervision, Validation, Investigation, Visualization, Methodology, Writing - original draft, Project administration, Writing - review and editing; Marta Kaczmarek, Investigation, Visualization, Methodology; Michał Mazur, Katarzyna Kuzmicz-Kowalska, Investigation, Methodology; Kamil Jastrzebski, Formal analysis, Visualization, Methodology; Marta Brewinska-Olchowik, Formal analysis, Visualization; Agata Kominek, Formal analysis; Katarzyna Piwocka, Resources, Funding acquisition; Marta Miaczynska, Conceptualization, Resources, Supervision, Funding acquisition, Writing - original draft, Writing - review and editing

## Author ORCIDs

Jaroslaw Cendrowski (iD) https://orcid.org/0000-0002-8579-7279
Marta Kaczmarek (iD) http://orcid.org/0000-0003-4939-6299
Michał Mazur (iD) http://orcid.org/0000-0001-5087-4409
Katarzyna Kuzmicz-Kowalska (iD) https://orcid.org/0000-0001-6146-1554
Kamil Jastrzebski (iD) https://orcid.org/0000-0001-6481-1759
Marta Brewinska-Olchowik (iD) https://orcid.org/0000-0002-2135-6220
Agata Kominek (iD) https://orcid.org/0000-0003-1567-9442
Katarzyna Piwocka (iD) https://orcid.org/0000-0001-6676-5282
Marta Miaczynska (iD) https://orcid.org/0000-0003-0031-5267

## Decision letter and Author response

Decision letter https://doi.org/10.7554/eLife.58504.sa1
Author response https://doi.org/10.7554/eLife.58504.sa2

# Additional files

### Supplementary files

• Supplementary file 1. Supplementary tables. **Tables 1 and 2**. List of proteins detected in BioID as proximal to BMP2K-L (1) or BMP2K-S (2) tagged with a mutant BirA biotin ligase (BirA*). Corresponding Gene symbols and Uniprot identifiers are provided. The list is ranked according to mean score between subtracted scores from N-terminally (N-tag) and C-terminally (C-tag) tagged baits. **Table 3**. List of shRNAs designed to deplete both (shBMP2K), or specific (BMP2K-L or BMP2K-S) BMP2K variants. Target nucleotide sequences as well as their locations on mRNA are provided. CDS – coding sequence, UTR – untranslated region. **Table 4**. List of gRNAs, non-targeting (gCtrl#1 and 2) or targeting *BMP2K* gene by CRISPR/Cas9 system (gBMP2K#1 and 2). When applicable, chromosomal position of base after cut by Cas9 as well as targeted DNA strand and location on gene are indicated. **Table 5**. List of primers used for assessing the levels of indicated human or mouse transcripts using qRT-PCR. Nucleotide sequences of both, forward and reverse primers are provided.

• Transparent reporting form

### Data availability

The mass spectrometry proteomics data have been deposited to the ProteomeXchange Consortium via the PRIDE partner repository with the dataset identifier PXD013542.

The following dataset was generated:

| Author(s) | Year | Dataset title | Dataset URL | Database and Identifier |
|---|---|---|---|---|
| Cendrowski J, Miaczynska M | 2019 | BMP2K is an inhibitor of erythroid differentiation that restricts endocytosis and SEC16A-dependent autophagy | http://www.ebi.ac.uk/pride/archive/projects/PXD013542 | PRIDE, PXD013542 |

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
