## [Decision Letter]

**Acceptance summary:**

In this work, the authors explore the roles of two forms (long, L, and short, S) of BMP2K that play a role in erythroid differentiation. BMP2K interacts with components of the COPII machinery that mediates ER export of secretory and membrane proteins, and also participates in autophagy. This suggests an intriguing regulatory mechanism for modulating cell fate by altering membrane trafficking events.

**Decision letter after peer review:**

Thank you for submitting your work entitled "Splicing variants of BMP2K, an erythroid-enriched endocytic kinase, coordinate SEC16A-dependent autophagy" for consideration by *eLife*. Your article has been reviewed by three peer reviewers, including Elizabeth A Miller as the Reviewing Editor and Reviewer #1, and the evaluation has been overseen by a Senior Editor.

Our decision has been reached after consultation between the reviewers. Based on these discussions and the individual reviews below, we regret to inform you that the work as submitted will not be considered for immediate publication in *eLife*. That said, there was considerable enthusiasm from all reviewers about the topic, specifically its relevance to erythroid differentiation. But all reviewers had significant reservations regarding various aspects of the study as presented. Of particular concern were the co-localization and immunofluorescence experiments that were difficult to discern and not quantified. All reviewers found the flow of logic difficult to follow, and a final model was unclear with respect to autophagy regulation and erythroid differentiation, and in particular to the role of Sec16 in this. All reviewers shared the opinion that a substantially revised and refocused manuscript that provides a clearer vision of the putative cross talk and its relevance to erythroid differentiation would be more compelling and suitable for *eLife*.

We hope that the attached reviewer comments, provided verbatim, will help guide you in terms of what was confusing or distracting in the current submission. These comments should serve as a guide for additional experiments that will be necessary to add clarity and ameliorate reviewers' concerns. We would be happy to consider a heavily revised submission, but also understand if you would prefer submission elsewhere for more timely publication.

Reviewer #1:

This manuscript from the Miaczynska lab describes potentially important insight into crosstalk between secretion, endocytic and autophagy pathways. Starting from the known endocytosis-associated protein, BMP2K, the authors use a combination of BioID, co-IP, co-localization and knock-down/knock-out studies to probe the multiple cellular pathways that BMP2K participates in. Some of the findings are strong, others less so, but the major concern is that causal links between any of the new effectors are lacking. The findings might better be presented as a focused report that highlights the new interactors and focuses on the most robust findings.

Essential revisions:

1) Overexpression experiments (Figure 1, Figure 2) clearly induce artifactual aggregates/structures. Although the authors are careful not to claim these represent endogenous structures, the findings are still somewhat overinterpreted. To my mind, the only thing these experiments demonstrate is that aggregates of BMP2K-L can sequester/recruit endocytic machinery.

2) The Sec16 co-IP experiment (Figure S2) is problematic: the no-GFP control IP clearly pulls down some BMP2K, raising the concern that the protein is sticky and will bind non-specifically. A GFP pulldown with an unrelated protein is probably a better control.

3) Some claims about co-localization and organelle morphology (Figure 3 and Figure 4D respectively) are difficult for me to see in the images presented. In the case of the Golgi morphology, is it possible that any differences are simply the result of increased ARFGAP? A control experiment overexpressing ARFGAP would address this. This problem highlights the causality issue in many interpretations.

4) I'm not sure exactly what the BFA experiment (Figure 5) tells us. BFA treatment has profound and pleiotropic effects, so again, the causal nature of what happens in these conditions is difficult to understand.

Reviewer #2:

The paper reports the characterization of molecular events leading to the transition for erythroid to red blood cells, which are devoid of nucleus as well as many organelles. The notion is that autophagy is activated during this transition leading to the clearance of many of these organelles.

The authors focus on two isoforms of the BMP2 kinase as this kinase appears to be upregulated during this transition.

The manuscript reports the differential localization and role of two isoforms of the, Short (S) and Long (L). BIoID has been performed with these two isoforms that bind different set of markers. When L is overexpressed in HeLa, it is found in aggregates and interacts with components of the endocytic machinery (that also localized in these aggregates) in line with the BioID. Conversely, S is more around or at the early secretory pathway and interact with ARFGAP and Sec16A.

The manuscript mostly focuses on S and Sec16A presented as a key player in autophagy.

They use two techniques. siRNA and CRISPR KO. shRNA of S revealed that Sec16A level is severely reduced but CRISPR KO does not reproduce any of this.

Taken together, the authors show that BMP2K is upregulated during the transition mentioned above yet it restricts autophagy. Furthermore, the reviewer is left with the notion that Sec16 is a negative regulator of autophagy and that BMP2K restricts autophagy, even though this kinase is upregulated during the transition and it is argued that autophagy should be stimulated.

At that stage, the manuscript is confusing and not publishable.

Essential revisions:

1) Sec16 has not been shown to be a positive player in autophagy. The paper Kundu, 2016 needs to be carefully read. Sec16 is phosphorylated by ULK1 but in basal conditions. In yeast, it indeed has been shown to be an autophagic player but this has not been confirmed in mammalian cells.

So Instead of presenting Sec16 as an established autophagic player, the authors should genuinely review all the evidences suggesting that it might be, and show that it is.

Furthermore, reading through the manuscript, the reviewer became increasingly confused as to whether the authors think/show that it is a positive or a negative factor for autophagy. This confusion therefore makes the reviewer weary of the title.

The Abstract is also confusing. What does functional cross talk with Sec16A mean?

2) I would disagree that the role of the early secretory pathway to autophagosome formation is still cloudy and badly understood. It is clearer and clearer especially since the publication of the article from Jeong, Cuervo and Pagano showing the switch from Sec23B being degraded in basal conditions to forming a "specific" coat for COPII vesicles dedicated to fuel the growth of the nascent autophagome.

In this regard, why are Sec24 and Sec23 not investigated in relation to S instead of Sec16 especially since Sec24 appears to be an interactor of this kinase. Sec24 level should be monitored.

3) In relation to this, what happens to general trafficking out of the ER through the secretory pathway upon manipulation of the 2 kinases isoforms (Oe, si RNA and CRISPR). This is not shown or even mentioned.

4) The co-localisation between S and Sec 16 is not convincing. Granted, some S is present at the compartments of the early secretory pathway upon OE in HeLa cells (Figure 2B). In the K562 cells, the cytoplasm is very small and would be cramped with most of the organelles. The reviewer doubts the colocalition is real even though the use of a relevant cell line is laudable.

Also puzzling is that overexpressed S co-immunoprecipitates with overexpressed GFP-Sec16 in HeLa cells. However, in the erythroid cell line K562, endogenous S does not co-precipitate Sec16 and Sec16 does not co-precipitate S.

5) However, it is not uncommon for a kinase to not localize near its substrate since binding is transient. Therefore, the emphasis should be on showing that Sec16 is phosphorylated by this kinase. Is the site known or can it be mapped? Is it conserved in species that do not have red blood cell?

6) the difference between shRNA and CRISPR KO is puzzling (to a certain extent) but removing Sec16 for BMP2K KO cells does not clarify the situation and leads to a mechanism. Either the shRNA has off targets and should not be used. Or Sec16 is not the right mechanism to focus upon.

7) The conclusion of the manuscript is that the two isoforms have antagonistic effects on the regulation of autophagy. S appears to inhibit it while L appears to activate it but overall, the enzyme restrict autophagy. Yet, both isoforms are upregulated in mouse fetal liver (so during the transition that is studied). How this make sense in a situation where autophagy should be stimulated to remove organelles from red blood cell cytoplasm?

8) The manuscript is complicated by the fact that it reports data on the endocytic markers that are hardly part of the story line and do not contribute to the clarity of the presentation. I would advise to remove the L part of the story with endocytosis and makes another article with it.

Reviewer #3:

This manuscript describes the membrane traffic functions of two splice variants of BMPK2 that influence pathways important for erythroid differentiation. There are several features of the study that are of interest to the broad readership of *eLife* and represent steps forward in understanding how membrane traffic pathways interact during cellular differentiation.

1) The membrane traffic role of this particular kinase, implicated in oncogenesis, has not been previously defined in detail.

2) The observation that two splice variants of the same gene regulate two different membrane traffic pathways (endocytosis and autophagy) that are both required for erythroid differentiation reveals the importance of balancing these two pathways for complex celllular functions.

3) This is another emerging example of cross-talk between endocytic and early secretory pathway players regulating specialized membrane traffic that leads to cell differentiation.

This study reports a number of different approaches to discovering the roles and functions of the BMPK2 splice variants and together the experimental results generally support the conclusions. The complexity of the findings make the story challenging to follow as a reader, since the data are revealed presumably in the order that the investigators followed their inquiry. However, this reviewer cannot obviously see another way to present the story, though the story is really about a dual regulator for membrane traffic pathways that influence erythroid differentiation.

A few issues need to be addressed in revision to support the interpretations of the data:

1) The initial cellular localization data shows that the two splice variants are localized differently, but quantification of overlap with the markers studied should be included for Figure 2-all panels.

2) The data in Figure S2C is much more convincing than the data in Figure 2C and D as an argument that mu2 phosphorylation is influenced by the L form. This should be included in the main figure.

3) It is clear that the downregulation of the different splice variants have different effects on distribution of secretory pathway markers (Figure 4D), but again quantification of co-localization changes should be provided.

4) There is some concern that the Tf uptake assay is not really measuring CME because there is no washing step after Tf binding (or at least this was not stated in the methods). So it seems that the 5 minute and the 40 minute time points both represent uptake of Tf that would be a combination of CME and bulk uptake. It has been reported that where CME is downregulated, bulk endocytosis is increased (eg with dynamin mutants), so both Tf and dextran uptake can increase together in these circumstances. Thus, it is not possible to conclude that BMP2K is a negative regulator of both types of endocytosis. Bulk uptake of both could increase because it is a positive regulator that is then removed. This doesn't change the story that both endocytosis and autophagy are mediated by the BMPK2 splice variants, but it would change the authors' discussion of how BMPK2 regulates the CME pathway.

[Editors’ note: further revisions were suggested prior to acceptance, as described below.]

Thank you for submitting your article "Splicing variation of BMP2K balances endocytosis, COPII trafficking and autophagy in erythroid cells" for consideration by *eLife*. Your article has been reviewed by three peer reviewers, including Elizabeth A Miller as the Reviewing Editor and Reviewer #1, and the evaluation has been overseen by Vivek Malhotra as the Senior Editor.

The reviewers have discussed the reviews with one another and the Reviewing Editor has drafted this decision to help you prepare a revised submission.

Essential revisions:

Reviewers had legitimate concerns about the claims, some of which are poorly substantiated, as well as some more technical concerns. After ample discussion, reviewers agreed to support a resubmission as a more focused Short Report rather than a Research Article. The Short Report format encompasses a novel important finding without the need for extensive mechanism. We believe your findings on the short and long forms of BMPK2, with their differential effects on COPII function fall into this category. In particular, the relevance of this to erythroid function broadens the general interest. To this end, we are making the following suggestions to streamline your findings and present only the most robustly supported findings. Where specific outcomes have not been demonstrated, you should tone down the conclusions. You should also acknowledge the limitations of the erythroid model you use with respect to full differentiation as noted by reviewer 3. Most importantly, you should not claim that secretion or COPII function has been impacted, since you don't measure secretion directly. Some aspects were repetitive (eg. showing both shRNA and gRNA KD/KO) and some of this duplication could be moved to supplementary figures, as suggested by reviewer 3. The discussion should be similarly streamlined.

One suggested revision plan:

Figure 1. A condensed composite of current Figure 1 and Figure 2 showing BMPK2 for the developmental trajectory and erythroid cell lines, and the relevance for erythroid differentiation. Here, the relevant comments of reviewers 2 and 3 should be taken into consideration.

(The mu-2 phosphorylation and endocytosis effects should be supplemental or omitted since causality here cannot be demonstrated. This aspect seems to be a side-line and disrupts the narrative flow and focus.)

Figure 2. Current Figure 4, showing mass spectrometry data and colocalization experiments (Eps15 as supplement). Again, note the technical concerns of reviewer 2 with respect to microscopy.

Figure 3. Effects of S and L forms on ERES (central versus peripheral), but tone down the interpretation on secretion unless experiments can be shown that quantify such an effect. Links to autophagy.

Figure 4. Link Sec16 data back to erythroid differentiation. Here, any additional data that support the function of autophagy vs. secretion switches in erythroid differentiation would be most helpful: S/L effects and Sec16 KD effects on erythroid differentiation.

Reviewer #1:

This manuscript investigates the role of BMPK2 splice isoforms in erythroid precursor cells, dissecting the role of the short and long forms in modulating endocytosis, exocytosis and autophagy. The data are of high quality, but the ms suffers from overinterpretation and the flow of logic could be vastly improved.

Specific concerns for discussion:

1) The relevance of the L/S ration in KO/KD experimetns versus simple abundance of either isoform isn't substantiated (Figure 2) so should be toned down with respect to causality.

2) Similarly, causality in regard to Tf internalization, mu-2 phosphorylation and erythroid development is not clear from the experiments, which makes conclusions difficult to draw. This section should be consolidated and simplified to spell out: observed effects are difficult to dissect because of changes in TfR abundance at the cell surface.

3) There are many situations where the findings are overstated (eg. Results section). In each case, the authors claim to have demonstrated regulation or control over COPII trafficking, when in reality all they have demonstrated is changes in abundance and different intracellular structures. In the absence of some measure of impact on secretion per se, this should be toned down.

4) Related to this problem of overstating conclusions, it's very muddy in the narrative as presented what the different COPII-positive structures represent (ERES vs autophagic sites) and how the different splice KDs impact these. Being more clear up front about what the model is would help with the flow of logic.

5) This would require some experimentation, but the over-riding question I had at the end was what impact the splice-specific and Sec16 KD had on erythroid differentiation. This seems to be a major conclusion that the authors want to make but the data were not there? I ended up making myself a table of the different findings to put it all together. This is something that would make the ms easier to follow.

Reviewer #2:

The manuscript by Cendrowski et al. studied the role of two isoforms of BMP2K and their differential effects on endocytosis, COPII trafficking and autophagy. They propose a model in which the differential biologic roles and expression of these two isoforms regulates erythroid maturation.

I noticed that this manuscript is a re-submission to *eLife*, but I see this work now for the first time. I assessed it independently of this, but had a look at how the authors had responded to the previous comments.

Overall, the topic of the study is highly interesting and the model the authors are proposing is intriguing. However, I was not convinced by many of the data and don't think that they support the claims of the authors. Too many interesting observations were made, that lack a mechanistic foundation, which strongly limits the impact of the work. In my view, this manuscript is not suitable for *eLife*.

Below are my main comments/concerns that are a technical as well as conceptual in nature.

1) Figure 1A,B: The blot shown does not really reflect the quantification. The problem is that the blot is a bit overexposed, making it hard to judge the bands properly.

2) Figure 1D: I agree that the L/S ratio changes, but both isoforms follow the same pattern, i.e. the increase from R0 to R3 and then drop in R5. Some changes in ratio are not detectable in the representative blot. For instance, the ratio at R1 and R5 is almost identical, yet the authors claim that there is a 2fold difference. Again, a better match between quantitation and representative blot is required to support the claim.

3) Figure 2: I agree that in panel B, the L/S ration in K562 cells is different. However, in panel A , the ratio is essential 1. The fact that this is "overexposed" as the authors state, is not sufficient to explain this discrepancy.

4) Figure 2C: Is the CRISPR-knockout incomplete, or are the remaining bands non-specific? The authors have not commented on this.

5) Conceptually: if you knockout BMP2K (i.e. the levels are 0), how can there be a ratio of any isoform of this gene. A ratio is only measurable if the knockout is not complete. Regardless of this conceptual problem, I don't think that L/S ratio is different between control and gRNA cells.

6) Figure 2: What sequence is the shRNA targeting? It is strange that the L/S ratio would change if the shRNA would target a sequence common to both isoforms. There is no explanation for this.

7) Figure 2H: Is the effect of g#1 really biologically significant? The difference in endocytosis of Tf or dextrane is less than 5%.

8) Figure 3: Why does the total level of TFRC increase when the whole BMP2K is depleted, but not when the individual isoforms are depleted? There is no mechanism for this observation.

9) Why do BMP2K-S depleted cells exhibit overall reduced endocytosis? Unless I am missing something, I find it hard to use the available data to explain this phenomenon.

10) Figure 4: the images are not of the best quality. I am well aware that cells such K562 cells are not as nice for microscopy studies compared to HeLa cells. However, the images are not really convincing. Because the authors are anyway using GFP-tagged L/S variants of BMP2K, I would suggest that they perform the colocalization studies in HeLa cells or any other cell line that is more suitable for imaging. To simply make the point that BMPK2 isoforms colocalize with Sec16A, HeLa cells should be sufficient. Single planes as well as maximum intensity projections should be shown.

11) It is important that the authors describe the sectioning (how many confocal planes and what section thickness). This is important, because the authors are using 40x objectives with NA1.1 or 1.3 and the information is required to judge the imaging procedure.

12) Figure 5: the authors try to make conclusions about ERES and COPII vesicles. This is absolutely not supported by data. The peripheral elements that the authors call "vesicles" are most certainly ERES. They are far too big to be COPII vesicles. Such vesicles would be maximally 80 nm in diameter, which is unlikely to be detectable using the staining protocols/methods that the authors are using. I see that a 40x objective with an NA of 1.3 was used. In addition, the authors used a point scanning confocal microscope. The quantum efficiency of standard PMT detectors for such microscopes is usually below 50%, making it unlikely to be able to detect vesicles.

13) Figure 5: The only method to determine whether any protein regulates COPII trafficking is to actually measure trafficking (i.e. a budding assay or a RUSH assay). The number of ERES does not necessarily correlate with the extent of trafficking defect. Thus, the conclusion that the authors are making are not supported by the data.

14) The finding that Sec16A depletion stimulates bulk autophagy is a highly surprising and begs for a mechanism.

15) Figure 6: the claim that BMP2K regulates production of COPII vesicles is not supported by the data at all (see comment above).

16) How does BMP2K-S regulate Sec16A protein stability? There is no mechanism. No CHX chase is performed. Is the degradation proteosomal or lysosomal?

17) The claim that BMP2K regulates the trafficking of Sec24B vesicles is not supported by the data.

Reviewer #3:

The manuscript by Miaczynska et al., is extensive study of the roles of two splice variants of BMP2K in erythroid cells (short and long isoforms). The work has strikingly moved the roles of the protein away from the suspected role in endocytosis and into ER to Golgi trafficking and also autophagy. The work initially starts out in mouse fetal liver cells and then moves to a more "simplified" model in K562 erythroleukemic cells. The premise that they go someway to prove is that the 2 BMPK isoforms have opposite functions- which is reinforced by the single knockdown data. The L form promotes COPII assembly and stimulates autophagy and the short form has an inhibitory role. These two processes are essential for erythroid differentiation.

The manuscript is a lengthy one, with knockdown experiments to determine effects of the different isoforms and also proximity labelling studies to identify interaction partners. Followed by lots of knockdown experiments to tease out the role. These data develop a significant case for BMP2K to have an important role in COPII coat formation and potentially autophagy. However, some of the data is extrapolated from a cell line that does not differentiate that well and so is not full proof for such a role in differentiation. This does not negate the worthiness of the article and I still judge it publishable in *eLife*, just means that the claims that are made need to be reinforced with statements such as this could be explored further in a erythroid differentiation systems or knockout mice experiments. I found the discussion poorly focused and would recommend an additional edit with a tight, organised structure that covers the pertinent points. That other related kinases may also have such a dual function is a nice point to make but perhaps could be made at the end not at the start of the discussion as its speculation. There is also no mention of how the short form could have a inhibitory effect. Do the authors think its because it lacks the c terminus and so competes with the long form?

Essential revisions:

1) The title and aspects of the paper need to remove all mention of COPII trafficking as this reviewer is not convinced this has been investigated directly. COPII formation has been explored here and cargo loading indirectly. No actual trafficking measurements. Sorry picky point.

2) Figure 2; it’s important to note in the text that the secretory pathway is lost during the late stages of differentiation so the drop in BMP2K at the late stages may not have any affect at this point as nearly everything that is not in a reticulocyte is being down regulated and is actively degraded. See for example Satchwell et al., 2013 for the human expression of various secretory components and before that paper there are multiple EM papers which show the pathway disappears. This alteration would not be seen in the K562 system. So the loss of the BMP2K protein expression seen in mice fetal liver and the ratio at the end may not matter. This point impacts on the second panel for schematic in Figure 8. It would be better to focus on abundance of each isoform relative to each other for the part of differentiation that there is a secretory pathway.

3) Where possible L/S ratio should still be calculated and provided when the proteins are depleted. I do think that all the depletion experiments are very repetitive. Could the authors consider some of this data being put in supplemental?

4) As mentioned at the start more in-depth studies of erythropoiesis are needed to fit the model and the suggestions. So instances of over interpretation and extrapolation should have a health warning attached or at least be said but then say this would need to be established in future work.

5) I am loath to add extra work but potential role of BMP2K in erythroid differentiation was studied in K562s induced to differentiate with hemin and this was only carried out when both isoforms where KD simultaneously (Figure 2 and S2). Did the authors conduct a KD separately of either L or S in K562 and if so, what was the effect? What about KD of Sec16? I feel I am missing some key experiments that add evidence in this simpler system. There is also the classic rescue experiment that we are lacking (overexpressing the L and S forms to alter the ration would be possible). Was this tried in K562? Or just overexpressing the isoforms and looking at the effects? Were any of these attempted?

---

## [Author Response]

[Editors’ note: the authors resubmitted a revised version of the paper for consideration. What follows is the authors’ response to the first round of review.]

Reviewer #1:This manuscript from the Miaczynska lab describes potentially important insight into crosstalk between secretion, endocytic and autophagy pathways. Starting from the known endocytosis-associated protein, BMP2K, the authors use a combination of BioID, co-IP, co-localization and knock-down/knock-out studies to probe the multiple cellular pathways that BMP2K participates in. Some of the findings are strong, others less so, but the major concern is that causal links between any of the new effectors are lacking. The findings might better be presented as a focused report that highlights the new interactors and focuses on the most robust findings.Essential revisions:1) Overexpression experiments (Figure 1, Figure 2) clearly induce artifactual aggregates/structures. Although the authors are careful not to claim these represent endogenous structures, the findings are still somewhat overinterpreted. To my mind, the only thing these experiments demonstrate is that aggregates of BMP2K-L can sequester/recruit endocytic machinery.

We agree with the reviewer. We believe that the formation of aggregates may reflect some biochemical properties of BMP2K-L. However, as indeed we have too little data for a proper interpretation, we decided to remove all results of BMP2K overexpression in HeLa cells (former Figure 2 and S2) and to focus only on results obtained in K562 cells.

2) The Sec16 co-IP experiment (Figure S2) is problematic: the no-GFP control IP clearly pulls down some BMP2K, raising the concern that the protein is sticky and will bind non-specifically. A GFP pulldown with an unrelated protein is probably a better control.

We thank the reviewer for pointing this out. First, we apologize for the mistake in the initial manuscript, where we named the control samples as no-GFP control (former Figure S2). In fact, we used a control plasmid overexpressing EGFP alone in the EGFP-SEC16A IP experiment in HEK293 cells (now shown in Figure 2B). Second, we have been indeed aware of BMP2K-S being sticky to the agarose beads. The presented analysis involves an already optimized co-IP protocol, which includes washes in high salt concentration to reduce the unspecific binding. Despite that, some unspecific signal is still observed and we admit it in the manuscript. In the revised version, to make sure that our results hold true, we reproduced the co-IP experiment several times and performed quantification from 3 independent repetitions (Figure 2B). This also allowed us to conclude that both BMP2K isoforms, and not only the S variant, may interact with SEC16A.

3) Some claims about co-localization and organelle morphology (Figure 3 and Figure 4D respectively) are difficult for me to see in the images presented. In the case of the Golgi morphology, is it possible that any differences are simply the result of increased ARFGAP? A control experiment overexpressing ARFGAP would address this. This problem highlights the causality issue in many interpretations.

Aiming at giving our manuscript a better focus on the most robust findings as suggested by the reviewer, we removed all data questioned in this point and replaced them with new results that provide more clear-cut information. The main change is that the revised manuscript focuses now on the function of SEC16A rather than of ARFGAP1. We recognize that our data regarding ARFGAP1 are promising but, as we realized thanks to the reviewers comments, they require a separate study.

We removed the analysis regarding ARFGAP1 and Golgi morphology shown in former Figure 4D and replaced them with the quantified IF data regarding SEC16A morphology and the distribution of COPII markers (Figure 3, Figure 3—figure supplement 1). Due to the shifted focus of the manuscript, we did not perform the control experiment overexpressing ARFGAP1 proposed by the reviewer. Instead, as a control experiment we performed depletion of SEC16A (former Figure 5).

On another note, we removed the IF analysis from former Figure 3 because we discovered that the mouse monoclonal antibody used to detect endogenous BMP2K inefficiently recognizes BMP2K-S in microscopy (shown in Figure 2—figure supplement 1), in contrast to immunoblotting where it recognizes both isoforms (see also response to reviewer 2, point 4).

4) I'm not sure exactly what the BFA experiment (Figure 5) tells us. BFA treatment has profound and pleiotropic effects, so again, the causal nature of what happens in these conditions is difficult to understand.

The reviewer is right and we removed the data of BFA experiments from the manuscript. Instead, we present a more detailed analysis of SEC16A function in K562 cells and its modulation by BMP2K-L and -S variants (Figure 3 and Figure 3—figure supplement 1).

Reviewer #2:The paper reports the characterization of molecular events leading to the transition for erythroid to red blood cells, which are devoid of nucleus as well as many organelles. The notion is that autophagy is activated during this transition leading to the clearance of many of these organelles.The authors focus on two isoforms of the BMP2 kinase as this kinase appears to be upregulated during this transition.The manuscript reports the differential localization and role of two isoforms of the, Short (S) and Long (L). BIoID has been performed with these two isoforms that bind different set of markers. When L is overexpressed in HeLa, it is found in aggregates and interacts with components of the endocytic machinery (that also localized in these aggregates) in line with the BioID. Conversely, S is more around or at the early secretory pathway and interact with ARFGAP and Sec16A.The manuscript mostly focuses on S and Sec16A presented as a key player in autophagy.They use two techniques. siRNA and CRISPR KO. shRNA of S revealed that Sec16A level is severely reduced but CRISPR KO does not reproduce any of this.Taken together, the authors show that BMP2K is upregulated during the transition mentioned above yet it restricts autophagy. Furthermore, the reviewer is left with the notion that Sec16 is a negative regulator of autophagy and that BMP2K restricts autophagy, even though this kinase is upregulated during the transition and it is argued that autophagy should be stimulated.At that stage, the manuscript is confusing and not publishable.Essential revisions:1) Sec16 has not been shown to be a positive player in autophagy. The paper Kundu, 2016 needs to be carefully read. Sec16 is phosphorylated by ULK1 but in basal conditions. In yeast, it indeed has been shown to be an autophagic player but this has not been confirmed in mammalian cells.So Instead of presenting Sec16 as an established autophagic player, the authors should genuinely review all the evidences suggesting that it might be, and show that it is.Furthermore, reading through the manuscript, the reviewer became increasingly confused as to whether the authors think/show that it is a positive or a negative factor for autophagy. This confusion therefore makes the reviewer weary of the title.

We are grateful to the reviewer for this insightful comment. Not to rely on unconfirmed assumptions regarding the role of SEC16A, we performed additional experiments to address the involvement of SEC16A in basal autophagy of K562 cells (Former Figure 5D-G). By shRNA and CRISPR/Cas9-mediated gene silencing we find that SEC16A is not required for basal autophagic degradation in these cells. On the contrary, we observed elevated autophagic flux in the absence of this protein. This discovery greatly impacted the interpretation of our further results. Namely, we found that in K562 cells, SEC16A is required for the production of juxtanuclear COPII vesicles but not of COPII assemblies dispersed outside of the juxtanuclear compartment (Figure 3D-E). Conversely, the absence of SEC16A increases recruitment of SEC31A to the dispersed structures. This suggests that SEC16A could inhibit COPII outer cage formation as demonstrated so far only for yeast Sec16. Importantly, we further show that SEC31A recruitment at SEC16A-positive ERES is regulated by BMP2K variants (Figure 3C-E, Figure 3—figure supplement 1).

The Abstract is also confusing. What does functional cross talk with Sec16A mean?

We removed the confusing sentence about functional cross talk with SEC16A from the abstract (and references to it elsewhere in the text) and replaced it with more precise conclusions regarding the role of BMP2K-L and -S in the regulation of SEC16A-dependent functions.

2) I would disagree that the role of the early secretory pathway to autophagosome formation is still cloudy and badly understood. It is clearer and clearer especially since the publication of the article from Jeong, Cuervo and Pagano showing the switch from Sec23B being degraded in basal conditions to forming a "specific" coat for COPII vesicles dedicated to fuel the growth of the nascent autophagome. In this regard, why are Sec24 and Sec23 not investigated in relation to S instead of Sec16 especially since Sec24 appears to be an interactor of this kinase. Sec24 level should be monitored.

We now provided more accurate information in the introduction section and analyzed SEC24B protein levels (Figure 3A) as well as its intracellular localization (Figure 3—figure supplement 1) in the absence of either of the BMP2K variants. The levels of SEC24B are largely unaffected, however the intracellular distribution of SEC24B is altered upon BMP2K-L or BMP2K-S depletion due to their effects on SEC16A-dependent trafficking.

3) In relation to this, what happens to general trafficking out of the ER through the secretory pathway upon manipulation of the 2 kinases isoforms (Oe, si RNA and CRISPR). This is not shown or even mentioned.

With respect to this valid point, we included new pieces of data showing the effects of depleting BMP2K variants on COPII distribution and the amount of SEC31A recruited to COPII structures (Figure 3C-E, Figure 3—figure supplement 1). We observe that both kinases markedly affect COPII production but in an opposite manner. BMP2K-L promotes while S restricts generation of COPII vesicles. Specifically, depletion of BMP2K-S leads to accumulation of SEC31A on SEC24B-positive COPII assemblies in a manner proportional to the amount of BMP2K-L left in the cell. Hence, cells lacking all BMP2K variants but with preserved high L/S ratio show moderately elevated COPII production.

Following the reviewers suggestion to provide a more focused manuscript, we decided to elaborate the role of BMP2K variants in autophagy. Thus, we did not analyze secretion of any model cargo. However, we observed changes in Tf receptor abundance on the plasma membrane (Figure 3D, open bars) that may result from altered COPII-mediated trafficking of Tf receptors, which we comment on in the Discussion section.

During the revision of our manuscript, we have been made aware of a report which provides additional rationale to address the role of BMP2K in autophagy. While studying an unrelated subject, Potts et al., 2013 fished out BMP2K as a putative regulator of autophagy and possibly mitophagy, also in erythroid cells. However, this study did not consider BMP2K as a putative membrane trafficking regulator or report its mechanisms of action. The involvement of particular BMP2K splicing variants was also not addressed. Potts et al. found BMP2K to potentially stimulate LC3-dependent autophagy. Extending this finding we show that it is the BMP2K-L variant that stimulates autophagy but in a manner counteracted by BMP2K-S. We further propose that this regulation may involve SEC16A-dependent COPII trafficking.

4) The co-localisation between S and Sec 16 is not convincing. Granted, some S is present at the compartments of the early secretory pathway upon OE in HeLa cells (Figure 2B). In the K562 cells, the cytoplasm is very small and would be cramped with most of the organelles. The reviewer doubts the colocalition is real even though the use of a relevant cell line is laudable.

We agree with the reviewer that in cells with small cytoplasm, such as K562, the organelles are so crowded that the observed colocalization may not reflect actual interaction of proteins. Nevertheless, what can be clearly distinguished is whether a given protein is enriched in particular cell regions, such as the plasma membrane region or the juxtanuclear region. Being aware that the analysis of overexpressed EGFP-tagged proteins does not allow drawing firm conclusions about protein associations, we use it only to complement the data from the BioID analysis. In the revised manuscript we show that we cannot rely on the monoclonal anti-BMP2K antibody in the IF analysis (Figure 2—figure supplement 2A,B), thus we removed such data from former Figure 3. The analysis of intracellular distribution of EGFP-tagged proteins overexpressed in K562 cells (Figure 2B and Figure 2—figure supplement 2C) is consistent with the co-IP with SEC16A in HEK293 cells (Figure 2B), that not only BMP2K-S but also BMP2K-L may have functions related to early secretory pathway. In agreement with this, we further show that indeed both variants regulate SEC16A function.

Also, puzzling is that overexpressed S co-immunoprecipitates with overexpressed GFP-Sec16 in HeLa cells. However, in the erythroid cell line K562, endogenous S does not co-precipitate Sec16 and Sec16 does not co-precipitate S.

We would like to clarify that in the previous manuscript we did not show data or claim in the text that endogenous BMP2K-S does not co-precipitate SEC16A and vice versa.

We agree with the reviewer about the importance of co-immunoprecipitation between the endogenous proteins as an approach confirming that interactions occur in the cell type of interest. However, this method is limited by the availability of antibodies. Both anti-BMP2K and anti-SEC16A antibodies are not suitable for efficient immunoprecipitation of endogenous proteins. Although we cannot confirm a physical interaction between BMP2K variants and SEC16A in K562 cells, we show a functional interaction, wherein both L and S BMP2K variants regulate SEC16A-dependent COPII production. Collectively, given (a) the detection of SEC16A as a protein proximal to BMP2K-S in BioID, (b) positive validation by co-IP in HEK293 cells and colocalization of EGFP-BMP2K variants in K562 cells as well as (c) the functional interaction, it is very likely that at least one BMP2K isoform interacts directly with SEC16A.

5) However, it is not uncommon for a kinase to not localize near its substrate since binding is transient. Therefore, the emphasis should be on showing that Sec16 is phosphorylated by this kinase. Is the site known or can it be mapped? Is it conserved in species that do not have red blood cell?

This is a very valid comment, which we wished to address. As we show in the Author response image 1, human SEC16A contains a Thr1045 residue, whose surrounding amino acid sequence is similar to the sites in μ2 and NUMB described as phosphorylated by AAK1 (Figure for the reviewers, panel A). This threonine is preceded by a sequence that matches a consensus for this family of kinases (Sorensen and Conner, 2008), namely φXXQXT (where φ is a hydrophobic residue and X is any amino acid). This putative phosphorylation site is highly conserved among vertebrates (Author response image 1 panel B) and we could not find a similar motif in the Sec16 sequences of *D. melanogaster*, *C. elegans* or *S. cerevisiae* (not shown). Hence, this site is not present in organisms that do not produce red blood cells.

We performed a preliminary analysis to test whether the changes in SEC16A protein levels observed by us upon depletion of BMP2K variants could result from altered phosphorylation by BMP2K. To this end we treated K562 cells with a pharmacological inhibitor of AAK1 and BMP2K (LP935509). It efficiently inhibited μ2 phosphorylation (Author response image 1 panel C), that as we show in our manuscript, is controlled primarily by BMP2K-L in K562 cells. Interestingly, this compound modestly upregulated SEC16A levels similarly to depletion of BMP2K-L (Author response image 1 panel C). This suggests that elevated SEC16A levels due to BMP2K-L depletion could result from reduced phosphorylation of SEC16A by BMP2K-L.

To test this, we mutated the T1045 site to alanine (TA) or glutamic acid (TE) and overexpressed the EGFP-tagged constructs in K562 cells (Figure Author response image 1 panel D-E). We observed that similarly to endogenous SEC16A, the overexpressed WT EGFP-SEC16A was upregulated upon shBMP2K-L and upon the LP inhibitor treatment. The same occurred for the TA mutant, however this upregulation did not occur for the TE mutant (Author response image 1 panel D-E). No upregulation of the TE mutant is a promising result, however the fact that levels of the TA mutant are regulated in the same way as those of WT does not allow drawing final conclusions from this experiment. It is however possible that the presence of a relatively big EGFP tag interferes with this analysis. We believe that very likely at least one of BMP2K variants phosphorylates SEC16A on T1045, however proving this properly will require a lot of time and substantial effort which we wish to devote in a follow-up study. We anticipate a complex scenario where BMP2K-L transiently interacts with SEC16A and phosphorylates T1045, while BMP2K-S stably binds SEC16A and alters either phosphorylation itself or the functional outcome of T1045 phosphorylation.

**Author response image 1. sa2fig1:** (A) Graphical comparison of amino acid sequences adjacent to Thr156 in μ2 and Thr102 in NUMB (both known to be phosphorylated in human cells by AAK1, a BMP2K homologue) and Thr1045 in SEC16A, that we identify as a candidate BMP2K phosphorylation site. The color code indicates that two upstream residues of SEC16A Thr1045 are identical to the respective residues in μ2, while two downstream amino acids are identical to those in NUMB. (B) Alignment of ~60 amino acids surrounding Thr1045 in human SEC16A and in its homologues from selected vertebrates. Green color indicates residues shown in A, grey color indicates a hydrophobic residue (φ) five amino acids upstream of the putative phosphorylation site. (C) Western blots showing the effect of increasing concentrations of LP935509, AAK1/BMP2K inhibitor, on the levels of endogenous SEC16A and phosphorylated μ2 in K562 cells. (D) Western blots showing the effects of control shRNA (Ctr), shBMP2K-L (L) or shBMP2K-S (S) on the levels of ectopically expressed EGFP-SEC16A either WT or with Thr1045 mutated to Ala (TA) or Glu (TE) in K562 cells. (E) Western blots showing the effects of the LP inhibitor on the levels of ectopically expressed WT or mutant EGFP-SEC16A in K562 cells.

6) the difference between shRNA and CRISPR KO is puzzling (to a certain extent) but removing Sec16 for BMP2K KO cells does not clarify the situation and leads to a mechanism ( see below point 11.3 and 13). Either the shRNA has off targets and should not be used. Or Sec16 is not the right mechanism to focus upon.

We agree that removing both SEC16A and BMP2K using the CRISPR/Cas9 approach is not suitable to clarify the interplay between SEC16A and BMP2K variants. As we observed that induction of erythroid differentiation is associated with reduced levels of SEC16A, there is no good rationale for co-depleting this protein together with BMP2K. We removed these data from the manuscript.

7) The conclusion of the manuscript is that the two isoforms have antagonistic effects on the regulation of autophagy. S appears to inhibit it while L appears to activate it but overall, the enzyme restrict autophagy. Yet, both isoforms are upregulated in mouse fetal liver (so during the transition that is studied). How this make sense in a situation where autophagy should be stimulated to remove organelles from red blood cell cytoplasm?

Inspired by this very relevant comment, we investigated in more detail the changes in protein levels of the two variants during mouse erythroblast differentiation. In addition to analyzing various differentiation time points (Figure 1A-B and Figure 1—figure supplement 3A), we assessed variant abundances in cells at particular differentiation stages, isolated by FACS (Figure 1C-D). We observe that although initially upregulated, the levels of both variants are reduced in late differentiation stages (erythroid maturation). Moreover, erythroid maturation is associated with increasing abundance of L over S (i.e. increasing L/S ratio) even at final stages. We validated these findings in K562 cells (former Figure 2) where depletion of all variants by shRNA (strongly promoting erythroid differentiation) is characterized by a strong increase in the L/S ratio while their depletion by CRISPR/Cas9 (modestly promoting differentiation) is associated with a modest increase in the L/S ratio. We further show that the two variants constitute the BMP2K-L/S regulatory system, wherein BMP2K-L promotes while BMP2K-S inhibits the intracellular processes important for erythroid differentiation, such as COPII trafficking (Figure 3, Figure 3—figure supplement 1) and autophagy (Figure 4D-F).

As we write in the Discussion section, our interpretation is that BMP2K-L is upregulated during erythropoiesis to promote erythroid maturation, while BMP2K-S inhibits this function (possibly as a negative feedback regulator). Therefore, at later stages of differentiation, upon reducing BMP2K-S levels below a certain threshold, BMP2K-L can exert its pro-maturation effects. However, as these are splicing variants, they should be (and appear to be) co-regulated on the transcriptional level, hence co-reduced upon maturation. We believe that the L/S ratio must be fine-tuned by additional mechanisms, which remain to be addressed. This fine-tuning may possibly allow adapting the erythropoiesis rate to altered demand for red blood cell production.

8) The manuscript is complicated by the fact that it reports data on the endocytic markers that are hardly part of the story line and do not contribute to the clarity of the presentation. I would advise to remove the L part of the story with endocytosis and makes another article with it.

We thank the reviewer for this suggestion. As the revised version is indeed more focused on the regulation of SEC16A function by BMP2K variants, for the purpose of clarity, we removed the analysis of endocytic markers in HeLa and HEK293 cells. However, we still show the involvement of BMP2K variants in endocytosis in K562 cells as a starting point in our analysis, before addressing other intracellular pathways (Former Figure 2G-H). Although based on literature data and our results, BMP2K is clearly an endocytic kinase we show that the role of its variants in regulation of endocytosis cannot explain their involvement in erythroid differentiation. We did not find evidence that the two variants would regulate endocytosis in an opposite manner as we did uncover for COPII trafficking and autophagy.

Although we recognize that the story-line would be more straightforward if focused only on one BMP2K variant, our data clearly show that both variants regulate SEC16A-dependent processes. In addition, as we find that BMP2K-S inhibits BMP2K-L-activated processes it is impossible to properly study one isoform in isolation from the other.

Reviewer #3:[…] A few issues need to be addressed in revision to support the interpretations of the data:1) The initial cellular localization data shows that the two splice variants are localized differently, but quantification of overlap with the markers studied should be included for Figure 2-all panels.

Please find our response below, along with point 2.

2) The data in Figure S2C is much more convincing than the data in Figure 2C and D as an argument that mu2 phosphorylation is influenced by the L form. This should be included in the main figure.

Both suggestions in (1) and (2) are absolutely right. However, to deliver a more focused message of the revised and restructured manuscript we decided to remove the data obtained in HeLa cells (former Figure 1B-C, and 2) and in HEK293 cells (former Figure S2C). Although they validate the results of the BioID analysis, they do not help to understand the role of BMP2K variants in the erythroid lineage.

3) It is clear that the downregulation of the different splice variants have different effects on distribution of secretory pathway markers (Figure 4D), but again quantification of co-localization changes should be provided.

The results in the former Figure 4D relate to ARFGAP1. In the revised version we decided to focus on SEC16A and COPII trafficking and for the sake of brevity we left out all data concerning an interesting link between BMP2K and ARFGAP1, to be elaborated in a follow-up study. Hence, these results have been replaced with analyses that focus on SEC16A function (Figure 3, Figure 3—figure supplement 1B,D), to which we have provided proper quantification, as requested by the reviewer.

4) There is some concern that the Tf uptake assay is not really measuring CME because there is no washing step after Tf binding (or at least this was not stated in the methods). So it seems that the 5 minute and the 40 minute time points both represent uptake of Tf that would be a combination of CME and bulk uptake. It has been reported that where CME is downregulated, bulk endocytosis is increased (eg with dynamin mutants), so both Tf and dextran uptake can increase together in these circumstances. Thus, it is not possible to conclude that BMP2K is a negative regulator of both types of endocytosis. Bulk uptake of both could increase because it is a positive regulator that is then removed. This doesn't change the story that both endocytosis and autophagy are mediated by the BMPK2 splice variants, but it would change the authors' discussion of how BMPK2 regulates the CME pathway.

We appreciate this insightful and very helpful comment. Indeed, in the previous version of the manuscript we showed only continuous Tf uptake which may not reflect the regulation of CME. In the revised manuscript we include analysis of the pulse chase Tf uptake that, as suggested by the reviewer, involves a washing step after Tf binding (former Figure 3D-E). As we wished to put more focus on possible differential functions of the two BMP2K variants, we performed this assay not only in cells lacking all BMP2K variants (shBMP2K), but also for isoform-specific depletions, shBMP2K-L and shBMP2K-S. This allowed us (a) to discover that among the two variants it is BMP2K-L that promotes Tf CME efficiency, (b) to find that BMP2K variants may regulate Tf receptor delivery to the plasma membrane, and (c) to conclude that regulation of Tf endocytosis could not explain the role of BMP2K variants in erythroid differentiation.

In addition, following the reviewer’s suggestion, we mention the possibility of BMP2K variants regulating bulk endocytosis in the Discussion section.

[Editors’ note: what follows is the authors’ response to the second round of review.]

[…] One suggested revision plan:Figure 1. A condensed composite of current Figures1 and Figure 2 showing BMPK2 for the developmental trajectory and erythroid cell lines, and the relevance for erythroid differentiation. Here, the relevant comments of reviewers 2 and 3 should be taken into consideration.(The mu-2 phosphorylation and endocytosis effects should be supplemental or omitted since causality here cannot be demonstrated. This aspect seems to be a side-line and disrupts the narrative flow and focus.)Figure 2. Current Figure 4, showing mass spectrometry data and colocalization experiments (Eps15 as supplement). Again, note the technical concerns of reviewer 2 with respect to microscopy.Figure 3. Effects of S and L forms on ERES (central versus peripheral), but tone down the interpretation on secretion unless experiments can be shown that quantify such an effect. Links to autophagy.Figure 4. Link Sec16 data back to erythroid differentiation. Here, any additional data that support the function of autophagy vs. secretion switches in erythroid differentiation would be most helpful: S/L effects and Sec16 KD effects on erythroid differentiation.

We would like to thank the Senior Editor, the Reviewing Editor and the peer reviewers for providing us with a clear path towards publication of our study and their constructive comments.

Following the editorial decision and instructions we have streamlined the manuscript to fit it to the Short Report format. We have removed or shortened parts that disturbed the flow of logic and have narrowed down the story to four main figures, as outlined by the editors. We have also corrected the quality of some western blotting or immunofluorescence images as requested by the reviewers. The part regarding the effects of BMP2K gene silencing on endocytosis has been shortened and presented in the supplement. The description of potential differences between the two BMP2K splicing variants in regulation of μ2 phosphorylation or endocytosis has been omitted. Instead, as requested by the reviewers, we have added new results showing the effects of depleting single variants (L or S) on erythroid differentiation of K562 cells. We have also analyzed the effect of SEC16A depletion on differentiation (see Author response image 2). However, the results of this analysis are difficult to interpret (see response to reviewer #1 point 5 and reviewer #3 point 5). Considering this and the fact that our surprising findings on the role of SEC16A in regulation of autophagy require further mechanistic explanation (as pointed out by reviewer #2), we have not included the data on SEC16A depletion in the current manuscript. We plan to use these results as a starting point for the follow-up study. Finally, we have toned down far-reaching conclusions indicated by the reviewers and have pointed out which aspects of the study require further investigation. The title of the resubmitted report is "Splicing variation of BMP2K balances abundance of COPII assemblies and autophagic degradation in erythroid cells".

Reviewer #1:This manuscript investigates the role of BMPK2 splice isoforms in erythroid precursor cells, dissecting the role of the short and long forms in modulating endocytosis, exocytosis and autophagy. The data are of high quality, but the ms suffers from overinterpretation and the flow of logic could be vastly improved.Specific concerns for discussion:1) The relevance of the L/S ration in KO/KD experimetns versus simple abundance of either isoform isn't substantiated (Figure 2) so should be toned down with respect to causality.

In the revised report, we have toned down our interpretation and conclusions (Results section).

2) Similarly, causality in regard to Tf internalization, mu-2 phosphorylation and erythroid development is not clear from the experiments, which makes conclusions difficult to draw. This section should be consolidated and simplified to spell out: observed effects are difficult to dissect because of changes in TfR abundance at the cell surface.

As requested, we have consolidated and simplified this section. The results on Tf internalization are now presented in the supplement (Figure 1—figure supplement 3).

3) There are many situations where the findings are overstated (eg. Results section). In each case, the authors claim to have demonstrated regulation or control over COPII trafficking, when in reality all they have demonstrated is changes in abundance and different intracellular structures. In the absence of some measure of impact on secretion per se, this should be toned down.

In the revised report, we have toned down all conclusions related to COPII trafficking and we have commented that they require further verification (Results section).

4) Related to this problem of overstating conclusions, it's very muddy in the narrative as presented what the different COPII-positive structures represent (ERES vs autophagic sites) and how the different splice KDs impact these. Being more clear up front about what the model is would help with the flow of logic.

We have been more precise in describing the results concerning the different COPII-positive structures. Our data show a correlation between increased SEC31A abundance on dispersed COPII assemblies and autophagy or erythroid differentiation. To our knowledge, COPII structures outside of the juxtanuclear secretory compartment have not been studied before. We have pointed out in the Results section and discussion section that whether and how these structures are related to autophagy and/or differentiation requires further investigation.

5) This would require some experimentation, but the over-riding question I had at the end was what impact the splice-specific and Sec16 KD had on erythroid differentiation. This seems to be a major conclusion that the authors want to make but the data were not there? I ended up making myself a table of the different findings to put it all together. This is something that would make the ms easier to follow.

We have included new data showing that depletion of BMP2K-L partially impairs, while depletion of BMP2K-S to some extent promotes erythroid differentiation (Figure 4D,E). We have also analyzed the effect of shRNA-mediated SEC16A depletion (Author response image 2 panel A) on erythroid differentiation of K562 cells, however these data are difficult to interpret. shSEC16A#1 strongly elevates hemoglobin production under both basal and hemin-stimulated conditions, while shSEC16A#2 impairs hemoglobin production upon hemin (Author response image 2 panel B). Neither of these effects is associated with consistent changes in the expression of erythroid markers (Author response image 2 panel C). We believe that BMP2K variants might affect SEC16A involvement in differentiation by altering its function (possibly via phosphorylation) rather than by regulating its levels, that remains to be addressed in the future. As we also write in response to point 14 of reviewer #2, we have removed the data regarding the effect of SEC16A depletion on COPII abundance and autophagy. This part would make the manuscript too long for the Short Report format and, as pointed out by reviewer #2, contains surprising results that need to be substantiated by mechanistic evidence. We plan to address this in a follow-up paper for which the removed parts will be a starting point.

**Author response image 2. sa2fig2:** (A) Representative western blot showing the efficiency of SEC16A depletion using two single shRNAs. (B) Percentage of benzidine-positive control K562 cells or cells depleted of SEC16A, under basal growth conditions or after stimulation for 48 h with 20 μM hemin (n=4 for basal or n=3 for hemin +/- SEM). (C) Fold changes in mRNA levels of the indicated erythroid markers in control cells or in cells depleted of SEC16A (n=3 +/- SEM).

Reviewer #2:The manuscript by Cendrowski et al. studied the role of two isoforms of BMP2K and their differential effects on endocytosis, COPII trafficking and autophagy. They propose a model in which the differential biologic roles and expression of these two isoforms regulates erythroid maturation.I noticed that this manuscript is a re-submission to eLife, but I see this work now for the first time. I assessed it independently of this, but had a look at how the authors had responded to the previous comments.Overall, the topic of the study is highly interesting and the model the authors are proposing is intriguing. However, I was not convinced by many of the data and don't think that they support the claims of the authors. Too many interesting observations were made, that lack a mechanistic foundation, which strongly limits the impact of the work. In my view, this manuscript is not suitable for eLife.Below are my main comments/concerns that are a technical as well as conceptual in nature.1) Figure 1A,B: The blot shown does not really reflect the quantification. The problem is that the blot is a bit overexposed, making it hard to judge the bands properly.

In the revised report we present less exposed blots in Figure 1A that better reflect the quantification shown in Figure 1B.

2) Figure 1D: I agree that the L/S ratio changes, but both isoforms follow the same pattern, i.e. the increase from R0 to R3 and then drop in R5. Some changes in ratio are not detectable in the representative blot. For instance, the ratio at R1 and R5 is almost identical, yet the authors claim that there is a 2fold difference. Again, a better match between quantitation and representative blot is required to support the claim.

We thank the reviewer for pointing this out. For the revised manuscript, we have performed additional biological repeats of the analysis shown in Figure 1D (now n=5). This is a technically difficult experiment as it requires FACS-based isolation of 6 distinct populations from primary cells stimulated in vitro. Despite high variability between biological repeats, we managed to obtain more reliable results shown in the western blot that matches the densitometric quantification. Thanks to this, we could specify that the highest L/S ratio in these cultures is observed in the R3 population (>2) but is still high (~2) in R4 and R5.

3) Figure 2: I agree that in panel B, the L/S ration in K562 cells is different. However, in panel A, the ratio is essential 1. The fact that this is "overexposed" as the authors state, is not sufficient to explain this discrepancy.

We have provided a better blot now shown in current Figure 1E.

4) Figure 2C: Is the CRISPR-knockout incomplete, or are the remaining bands non-specific?The authors have not commented on this.

Within the time-frame of our analysis, the CRISPR/Cas9 approach in K562 cells does not lead to complete BMP2K knock-out although it efficiently reduces the levels of both variants (shown in Figure 1—figure supplement 1C). We have pointed this out in the Results section.

5) Conceptually: if you knockout BMP2K (i.e. the levels are 0), how can there be a ratio of any isoform of this gene. A ratio is only measurable if the knockout is not complete. Regardless of this conceptual problem, I don't think that L/S ratio is different between control and gRNA cells.

Indeed, as pointed out above, the knockout is not complete. To assess the L/S ratio we have performed quantification of multiple biological repetitions (n=5) of CRISPR/Cas9-mediated BMP2K depletion and we have obtained highly significant increase of the L/S ratio with respect to control cells. Moreover, we have provided a better western blot (short and long exposures) in the Figure 1—figure supplement 1C.

As compared to other cell types, cell of the erythroid lineage and K562 cells contain very high levels of BMP2K protein variants. It is thus possible that even upon efficient depletion, the remaining levels of either of the variants are high enough to affect erythroid differentiation.

6) Figure 2: What sequence is the shRNA targeting? It is strange that the L/S ratio would change if the shRNA would target a sequence common to both isoforms. There is no explanation for this.

The shRNA is targeting a sequence encoding part of the kinase domain, present in both variants. In current Supplementary file 1—table 3, we have provided the information regarding locations of target nucleotide sequences of shRNAs used in our study.

As we now write in the Results section, we have no clear explanation for changing L/S ratio upon shRNA. However, there are several possible explanations for this, including: (a) BMP2K-L protein could be more stable than BMP2K-S, (b) mRNA of BMP2K-L could be more abundant than that of -S and therefore more difficult to remove by RNAi, (c) mRNA for BMP2K-L could be located in a different cellular compartment (potentially due to local translation or mRNA transport) where it could be less accessible for shRNA. In order to keep the new report version concise, we did not include these speculations in the manuscript.

7) Figure 2H: Is the effect of g#1 really biologically significant? The difference in endocytosis of Tf or dextrane is less than 5%.

We agree with the reviewer that this effect of gBMP2K#1 is very weak, although it is one of several results showing the same tendency. Actually, such a weak effect fits with our general view that upregulated endocytosis upon BMP2K depletion does not induce erythroid differentiation but could rather be a secondary effect of induced differentiation.

8) Figure 3: Why does the total level of TFRC increase when the whole BMP2K is depleted, but not when the individual isoforms are depleted? There is no mechanism for this observation.

To improve the flow of logic, in the new report version, we have removed the results showing effects of BMP2K-L or -S depletion on total and cell surface abundance of TFRC.

9) Why do BMP2K-S depleted cells exhibit overall reduced endocytosis? Unless I am missing something, I find it hard to use the available data to explain this phenomenon.

We have shortened the description of results regarding TFRC levels and endocytosis and toned down our conclusions (subsection “The role of BMP2K in CME does not explain its involvement in erythroid differentiation”). The only conclusion that we now make is that the role of BMP2K variants in regulation of endocytosis cannot explain their involvement in erythroid differentiation.

10) Figure 4: the images are not of the best quality. I am well aware that cells such K562 cells are not as nice for microscopy studies compared to HeLa cells. However, the images are not really convincing. Because the authors are anyway using GFP-tagged L/S variants of BMP2K, I would suggest that they perform the colocalization studies in HeLa cells or any other cell line that is more suitable for imaging. To simply make the point that BMPK2 isoforms colocalize with Sec16A, HeLa cells should be sufficient. Single planes as well as maximum intensity projections should be shown.

We apologize for insufficient quality of pictures in the former Figure 4. Now we show better images in the current Figure 2C (more representative cells were chosen and improved imaging conditions were applied). In the initially submitted manuscript, we had provided immunofluorescence data in HeLa cells but the former reviewers advised us to remove these results, as ectopic expression of EGFP-BMP2K-L in HeLa led to formation of big aggregates, likely artifacts of overexpression. In addition, the analysis in yet another cell line disrupted the flow of logic of the manuscript. Therefore, we had been advised to show only immunofluorescence in K562 cells.

On a technical note, we performed the staining procedure in suspension and stained cells were immobilized on the microscopy plates in agarose. As a result, the imaged cells were not exactly at the same z-position. Therefore, single stacks were not suitable for analysis. As we have shortened the manuscript to the Short Report format there is no space for showing both single stack and projection images.

11) It is important that the authors describe the sectioning (how many confocal planes and what section thickness). This is important, because the authors are using 40x objectives with NA1.1 or 1.3 and the information is required to judge the imaging procedure.

For the confocal microcopy experiments, shown in current Figure 2C, Figure 2—figure supplement 2A, Figure 3C, and Figure 3—figure supplement 1C, we captured images from 7-8 sections with 1 μm interval in the z axis. This information is also included in the Materials and methods section.

12) Figure 5: the authors try to make conclusions about ERES and COPII vesicles. This is absolutely not supported by data. The peripheral elements that the authors call "vesicles" are most certainly ERES. They are far too big to be COPII vesicles. Such vesicles would be maximally 80 nm in diameter, which is unlikely to be detectable using the staining protocols/methods that the authors are using. I see that a 40x objective with an NA of 1.3 was used. In addition, the authors used a point scanning confocal microscope. The quantum efficiency of standard PMT detectors for such microscopes is usually below 50%, making it unlikely to be able to detect vesicles.

We agree with the reviewer and in the revised version we have refrained from calling the detected structures as “vesicles”. Instead we state in the Title, Abstract, and Results section that we observe effects on the distribution and abundance of COPII assemblies.

13) Figure 5: The only method to determine whether any protein regulates COPII trafficking is to actually measure trafficking (i.e. a budding assay or a RUSH assay). The number of ERES does not necessarily correlate with the extent of trafficking defect. Thus, the conclusion that the authors are making are not supported by the data.

We thank the reviewer for pointing this out. In the revised manuscript, we refrain from concluding that COPII trafficking is regulated. We only mention that whether BMP2K variants regulate COPII trafficking requires further investigation subsection “The BMP2K-L/-S system regulates abundance and distribution of COPII assemblies”).

14) The finding that Sec16A depletion stimulates bulk autophagy is a highly surprising and begs for a mechanism.

As we also write in the response to point 5 of reviewer #1, we have removed the data regarding depletion of SEC16A. We agree that such a surprising result requires investigation of an underlying mechanism, that we wish to perform in a follow-up study.

15) Figure 6: the claim that BMP2K regulates production of COPII vesicles is not supported by the data at all (see comment above).

We thank the reviewer for pointing this out. In the revised manuscript, we conclude that BMP2K variants regulate the abundance and distribution of COPII assemblies (page 9).

16) How does BMP2K-S regulate Sec16A protein stability? There is no mechanism. No CHX chase is performed. Is the degradation proteosomal or lysosomal?

These are very important questions that we have not addressed yet. We wish to focus on them in the follow-up study.

17) The claim that BMP2K regulates the trafficking of Sec24B vesicles is not supported by the data.

We agree with the reviewer. In the revised manuscript we use SEC24B only as a marker of COPII assemblies and the trafficking of SEC24B-positive vesicles is not addressed or mentioned in the text.

Reviewer #3:The manuscript by Miaczynska et al., is extensive study of the roles of two splice variants of BMP2K in erythroid cells (short and long isoforms). The work has strikingly moved the roles of the protein away from the suspected role in endocytosis and into ER to Golgi trafficking and also autophagy. The work initially starts out in mouse fetal liver cells and then moves to a more "simplified" model in K562 erythroleukemic cells. The premise that they go someway to prove is that the 2 BMPK isoforms have opposite functions- which is reinforced by the single knockdown data. The L form promotes COPII assembly and stimulates autophagy and the short form has an inhibitory role. These two processes are essential for erythroid differentiation.The manuscript is a lengthy one, with knockdown experiments to determine effects of the different isoforms and also proximity labelling studies to identify interaction partners. Followed by lots of knockdown experiments to tease out the role. These data develop a significant case for BMP2K to have an important role in COPII coat formation and potentially autophagy. However, some of the data is extrapolated from a cell line that does not differentiate that well and so is not full proof for such a role in differentiation. This does not negate the worthiness of the article and I still judge it publishable in eLife, just means that the claims that are made need to be reinforced with statements such as this could be explored further in a erythroid differentiation systems or knockout mice experiments. I found the discussion poorly focused and would recommend an additional edit with a tight, organised structure that covers the pertinent points. That other related kinases may also have such a dual function is a nice point to make but perhaps could be made at the end not at the start of the discussion as its speculation. There is also no mention of how the short form could have a inhibitory effect. Do the authors think its because it lacks the c terminus and so competes with the long form?

We thank the reviewer for these constructive comments. We have shortened and refocused the Discussion section, as suggested.

Regarding a possible mechanism by which BMP2K-S could have an inhibitory effect, it might involve some kind of competition between the two variants. However, as both variants have the kinase domain, the mechanism is likely more complex. As found in a high throughput proteomic study, SEC16A might interact with AP-2 and clathrin proteins (Hein et al., 2015). BMP2K-L has multiple AP-2 and clathrin binding sites in its C-terminus, absent from BMP2K-S. Therefore, association with CME proteins could possibly affect the functional interaction between BMP2K variants and SEC16A. However, to fit the short rShort Report format we have not included these speculations in the manuscript text.

Essential revisions:1) The title and aspects of the paper need to remove all mention of COPII trafficking as this reviewer is not convinced this has been investigated directly. COPII formation has been explored here and cargo loading indirectly. No actual trafficking measurements. Sorry picky point.

As requested by the reviewer, we have removed any references to COPII trafficking from the manuscript title, Abstract, subtitles and conclusions. We only mention that whether BMP2K variants regulate COPII trafficking requires further investigation (subsection “The BMP2K-L/-S system regulates abundance and distribution of COPII assemblies”).

2) Figure 2; it’s important to note in the text that the secretory pathway is lost during the late stages of differentiation so the drop in BMP2K at the late stages may not have any affect at this point as nearly everything that is not in a reticulocyte is being down regulated and is actively degraded. See for example Satchwell et al., 2013 for the human expression of various secretory components and before that paper there are multiple EM papers which show the pathway disappears. This alteration would not be seen in the K562 system. So the loss of the BMP2K protein expression seen in mice fetal liver and the ratio at the end may not matter. This point impacts on the second panel for schematic in Figure 8. It would be better to focus on abundance of each isoform relative to each other for the part of differentiation that there is a secretory pathway.

We have added the information about loss of the secretory pathway during the late stages of differentiation in the introduction of the revised manuscript, citing the relevant references. The reviewer’s comment touches upon a very interesting issue. On the one hand, the secretory pathway is required for rearrangement of PM of maturing erythroblasts and on the other it is eventually lost in reticulocytes. As described, among consecutive stages of mouse fetal liver erythroblast differentiation, reticulocytes are in the last, R5 stage (Zhang et al., 2003). We observe an increase in the L/S ratio already in earlier stages, being the highest in R3 when cells start to expose erythroid-specific markers (such as Ter-119) on their surfaces. Hence, BMP2K levels and the L/S ratio are high upon transition that would require activation of erythroid-specific secretion. Moreover, BMP2K levels are reduced with the high L/S ratio when cells begin to terminally differentiate (R4-R5), at the time of loss of the COPII machinery. We have briefly commented on this in the Discussion section of the revised manuscript. In turn, former Figure 8 with the final model schematic has been removed to shorten the manuscript to the Short Report format.

3) Where possible L/S ratio should still be calculated and provided when the proteins are depleted. I do think that all the depletion experiments are very repetitive. Could the authors consider some of this data being put in supplemental?

We agree that the depletion experiments are repetitive but provide additional verification of our observations. In order to adapt the manuscript to the Short Report format, we chose not to provide the L/S ratio at all times when depletions are shown. To avoid repetitive experiments in the main figures, we have moved the results of CRISPR/Cas9-mediated BMP2K depletion to the supplement (current Figure 1—figure supplement 3).

4) As mentioned at the start more in-depth studies of erythropoiesis are needed to fit the model and the suggestions. So instances of over interpretation and extrapolation should have a health warning attached or at least be said but then say this would need to be established in future work.

As requested by the reviewer, we have commented about the limitations of using K562 cells as the model of erythroid differentiation (mentioned in the Results section and the Discussion section).

5) I am loath to add extra work but potential role of BMP2K in erythroid differentiation was studied in K562s induced to differentiate with hemin and this was only carried out when both isoforms where KD simultaneously (Figure 2 and S2). Did the authors conduct a KD separately of either L or S in K562 and if so, what was the effect? What about KD of Sec16? I feel I am missing some key experiments that add evidence in this simpler system. There is also the classic rescue experiment that we are lacking (overexpressing the L and S forms to alter the ration would be possible). Was this tried in K562? Or just overexpressing the isoforms and looking at the effects? Were any of these attempted?

As requested also by the Editor and reviewer #1, we have analyzed the effect of depleting either L or S on erythroid differentiation of K562 cells (current Figure 4D,E). As we also write in response to point 5 of Reviewer #1, we observe that cells lacking BMP2K-L differentiate less potently, while those lacking BMP2K-S differentiate more potently. Interestingly, cells lacking BMP2K-S only do not behave exactly like cells lacking all variants. Thus, it is possible that both variants need to be reduced with an optimal L/S ratio to potently induce differentiation. We have also analyzed the effect of SEC16A depletion on K562 cell differentiation (Author response image 2 panels A-C). However, although shSEC16A#1 increased hemoglobin production in basal and hemin-stimulated conditions, shSEC16A#2 inhibited the induction of hemoglobin production upon hemin (Author response image 2 panel B). In addition, the two shRNAs caused changes in mRNA levels of erythroid markers that were not consistent with the effects on hemoglobin production ( Author response image 2 panel C). Hence, although we observed that SEC16A inhibits basal autophagic degradation (former Figure 5, currently omitted), we could not confirm that it restricts erythroid differentiation. It is possible that BMP2K variants differentially affect SEC16A function, which might be difficult to dissect by SEC16A depletion. So far, we have not tested the effects of overexpressing BMP2K variants in normal or BMP2K-depleted K562 cells on erythroid differentiation, though this is a very good suggestion. Given the laboratory work limitations in our institutions due to the current epidemiological situation, we could not attempt to perform such experiments during the revision. However, we plan to perform them in the future.